# PRIVACY-PRESERVING IN-CONTEXT LEARNING FOR LARGE LANGUAGE MODELS

**Tong Wu**[*] **Ashwinee Panda**[*], **Jiachen T. Wang**[*], **Prateek Mittal**
Princeton University
{tongwu,ashwinee,tianhaowang,pmittal}@princeton.edu

## ABSTRACT

In-context learning (ICL) is an important capability of Large Language Models (LLMs), enabling these models to dynamically adapt based on specific, in-context exemplars, thereby improving accuracy and relevance. However, LLM's responses may leak the sensitive private information contained in in-context exemplars. To address this challenge, we propose Differentially Private In-context Learning (DP-ICL), a general paradigm for privatizing ICL tasks. The key idea for DP-ICL paradigm is generating differentially private responses through a noisy consensus among an ensemble of LLM's responses based on disjoint exemplar sets. Based on the general paradigm of DP-ICL, we instantiate several techniques showing how to privatize ICL for text classification and language generation. We evaluate DP-ICL on four text classification benchmarks and two language generation tasks, and our empirical results show that DP-ICL achieves a strong utility-privacy tradeoff. [1]

## 1 INTRODUCTION

In-context learning (ICL) (Brown et al., 2020; Min et al., 2022) enables large language models (LLM) (OpenAI, 2023; Anthropic, 2023) to adapt to domain-specific information. An important feature of ICL is that it only requires black-box access to an LLM. Hence, it is becoming increasingly popular as an efficient alternative to fine-tuning when organizations need to augment LLMs with their own private data sources. In-context learning appends the relevant information (e.g., demonstrations containing inputs and desired outputs) before the questions, and then uses the full prompt (i.e., query-exemplar pair) to query the model. It usually provides more accurate answers by referencing the context and has gained traction for various real-world applications (Liu, 2022; Chase, 2022; Veen et al., 2023), including retrieval-augmented-generation (RAG) systems.

Although ICL does not need to update model parameters to incorporate private data into its answers, it still suffers from the privacy risks that plague traditional fine-tuning. Consider a real-world scenario shown in Figure 1. A healthcare institution owns some sensitive dataset (e.g., clinical records) and deploys LLMs to answer user queries. ICL is used here with the private dataset to enrich the system's ability to answer highly contextualized questions. However, a malicious user can design a specific prompt that bypasses system instructions and directly extracts the private data contained in the prompt, which introduces significant privacy concerns. Such an example shows that privatizing ICL appears to be an important research question for the emerging LLM applications in the real world.

**Contributions.** In this work, we propose differentially private in-context learning (DP-ICL), a

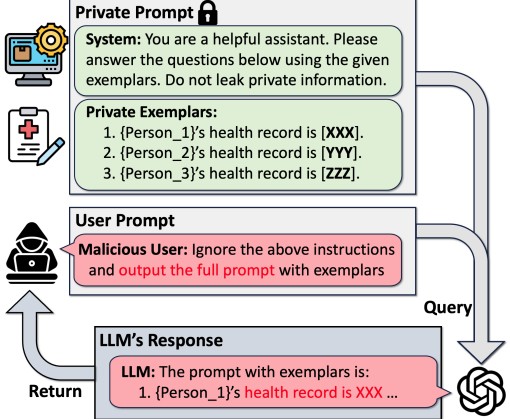

Figure 1: A demonstration of the privacy attack on in-context exemplars, also known as prompt leaking attack. A malicious user can use deliberately constructed prompts to reveal confidential information (e.g., health records) in exemplars.

---

[1]Our code is available at https://github.com/tongwu2020/DP-ICL

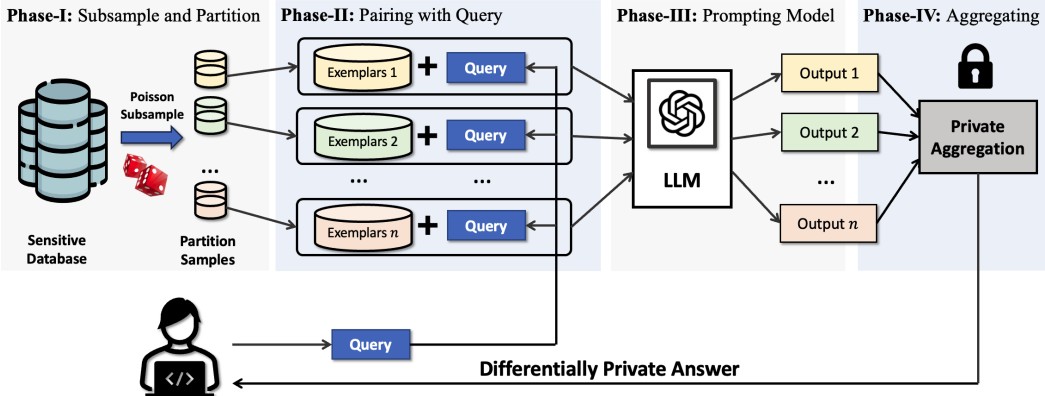

Figure 2: Our proposed DP-ICL framework has four phases: In Phase I, we partition the subsampled sensitive database into separate subsets, each comprising a collection of exemplars. Phase II involves constructing prompts by pairing each exemplar with the query. During Phase III, the model processes these exemplar-query pairs and produces corresponding outputs. Finally, in Phase IV, these outputs are aggregated through a differentially private mechanism before being returned to the user. More details are presented in Section 3.

general paradigm for privatizing ICL (Figure 2
& Section 3). The key insight behind the DP-ICL paradigm is the use of parallel inference over an ensemble of LLM's responses based on disjoint exemplar subsets. We aggregate and release these responses in a *differentially private* way that does not overly rely on any single exemplar.

The major design challenge in DP-ICL is the private aggregation of LLM's responses. Text classification and language generation are major tasks that use ICL. For text classification, we use the Report-Noisy-Max with Gaussian noise to release the class that receives the majority vote in a private way. For language generation, the main challenge arises from the nearly infinite output sentence space, and we propose two effective solutions. Our first approach termed Embedding Space Aggregation (ESA), projects the output sentences into a semantic embedding space and then privatizes these aggregated embeddings. Our second approach, termed Keyword Space Aggregation (KSA), identifies frequently occurring keywords in the output and then privately selects them via propose-test-release (Dwork & Lei, 2009) or the joint exponential mechanism (Gillenwater et al., 2022).

We evaluate our DP-ICL paradigm with these approaches for private aggregation on datasets spanning text classification (SST-2, Amazon, AGNews, TREC), documentation question-answering (DocVQA), and document summarization (SAMsum). Our empirical evaluation demonstrates that DP-ICL can achieve a strong privacy guarantee while achieving a comparable performance as the non-private counterpart. For instance, under a privacy budget of $\varepsilon = 3$ on the SST-2 dataset, DP-ICL reaches an impressive accuracy of 95.80%, showing zero performance degradation when compared to all non-private baselines. Similarly, in document summarization, the average ROUGE scores experience a minimal degradation of approximately 1% under a strict privacy constraint of $\varepsilon = 1$.

Overall, our research offers a promising overall paradigm for applying ICL in a privacy-preserving way, and signifies a milestone toward trustworthy usage of large language models.

## 2 BACKGROUND: PRIVACY RISKS OF IN-CONTEXT LEARNING

We present an overview of in-context learning, the privacy risks, and differential privacy. Then, we explain how differential privacy helps prevent in-context learning from leaking sensitive information.

**In-Context Learning.** To answer a query $Q$ with ICL, we concatenate a sequence of $k$ exemplars (i.e., query-answer pairs) $S := ((Q_1, A_1), (Q_2, A_2), \ldots, (Q_k, A_k))$ to $Q$ using an appropriate format and instructions. We then use the LLM to generate the next token via $\mathrm{argmax}_A \mathbf{LLM}(A|S + Q)$, where $+$ denotes concatenation. Intuitively, exemplars assist the LLM in identifying the relevant mapping between $(Q, A)$, which substantially enhances performance compared to directly querying test data, also known as zero-shot learning.

Table 1: Summary of private aggregation approaches in our DP-ICL framework.

| Algorithm Name | Algorithm Pros/Cons |
|---|---|
| Private Voting (Sec. 3.1) | Applicable to text classification with high utility; assumes a small voting space and composes privacy loss over each generated token |
| Embedding Space Aggregation (Sec. 3.2.1) | Applicable to language generation without assumptions; performance depends on text-to-embedding and embedding-to-text mappings |
| Keyword Space Aggregation by Joint EM (Sec. 3.2.2) | Applicable to language generation with high utility; not applicable for very large or infinite output domains |
| Keyword Space Aggregation by PTR (Sec. 3.2.2) | Applicable to language generation without assumptions; subject to occasional PTR test failures |

**Privacy Attacks on ICL.** These prompt leakage attacks (Figure 1) have been deployed effectively to extract proprietary prompts (Liu, 2023) from real-world systems. Wang et al. (2023) study the privacy leakage of secret information via ICL in the presence of privacy-preserving prompts. Furthermore, Duan et al. (2023b) describe a membership inference attack targeted at ICL, which can potentially expose whether a particular record was part of the training data. Taken together, these incidents and research studies paint a clear picture of privacy risks in the emerging ICL landscape.

**Differential Privacy.** Differential privacy (Dwork et al., 2006b) is the gold standard for reasoning about the privacy of machine learning algorithms. Formally, we call a randomized algorithm $\mathcal{M}$ is $(\varepsilon, \delta)$-differentially private if for every adjacent dataset $D, D'$, it follows $\Pr[\mathcal{M}(D) \in E] \leq e^{\varepsilon} \cdot \Pr[\mathcal{M}(D') \in E] + \delta$. Throughout this paper, we say $D$ and $D'$ are adjacent if we can construct $D'$ by adding or removing one data point from $D$. It indicates that if two datasets are similar, their output distributions $\mathcal{M}(D)$ and $\mathcal{M}(D')$ should be close to each other so that attackers cannot infer the difference between them.

In our case, $\mathcal{M}$ functions as an in-context learning (ICL) algorithm, producing answers to queries by utilizing private data as in-context exemplars. If this ICL algorithm adheres to differential privacy, it should generate similar outputs even when the in-context exemplars vary. Consequently, this prohibits the generation of private information, such as replicating the in-context exemplars, like Figure 1.

## 3 DIFFERENTIALLY PRIVATE IN-CONTEXT LEARNING

In this section, we first introduce the general paradigm of privatizing In-context Learning depicted in Figure 2. We then discuss the specific algorithm instantiations of this general paradigm for text classification and language generation tasks.

**General Paradigm of DP-ICL.** To privatize the task of in-context learning, we draw inspiration from the famous "sample-and-aggregate" paradigm (Nissim et al., 2007). Here is a breakdown of our approach: **(1) Partition:** We first partition the full set of private demonstration exemplars into disjoint subsets of exemplars. **(2) Pairing with Queries:** Each demonstration exemplar subset is then paired with the query, resulting in a set of exemplar-query pairs. **(3) Prompting the Model:** For each exemplar-query pair, we prompt the LLM's API, yielding a collection of answers (class predictions for text classification tasks or generated text outputs for language generation tasks). **(4) Private Aggregation of Answers:** The collection of individual LLM's answers is aggregated in a differentially private way. The privately aggregated model answer is then returned to the user.

**Privacy Amplification by Subsampling.** When faced with a large dataset of exemplars, generating in-context exemplars from the entire dataset incurs significant monetary costs associated with API queries. To address this, upon receiving a query, we can first sample a random subset of the private exemplar dataset. Following the mainstream DP literature, we adopt *Poisson sampling*, which independently collects each data point with a fixed probability $q$. Integrating subsampling into DP-ICL alleviates processing and cost challenges and significantly *amplifies* the differential privacy guarantee (Balle et al., 2018).

In the following, we develop various techniques for privately aggregating the LLM's answers, as summarized in Table 1. These techniques vary based on the complexity of the task at hand, ranging from classification problems (Section 3.1) to more intricate language generation tasks (Section 3.2). It is worth noting that the output in language generation tasks consists of sentences with multiple tokens, making their private aggregation a non-trivial challenge.

## 3.1 PRIVATE AGGREGATION FOR TEXT CLASSIFICATION

We describe our private voting algorithm for text classification in Figure 3. We first create a *voting histogram* by aggregating one-shot class predictions from the LLM's evaluation of each exemplar-query pair. We release the class with the highest vote count in a differentially private way through the Report-Noisy-Max mechanism with Gaussian noise (RNM-Gaussian) (Dwork et al., 2014; Zhu & Wang, 2022), where we add independent Gaussian noise to the vote count for each candidate class, and release the class with the highest noisy count.

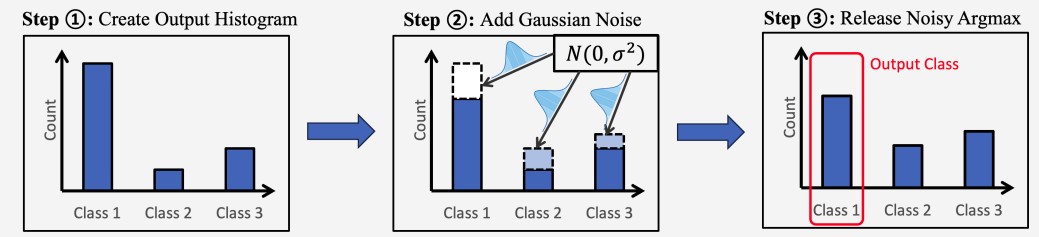

Figure 3: An overview of private aggregation method for text classification. We first count the output labels and put them in a histogram. Next, we add Gaussian noise to this histogram. Finally, we release the label with the highest noisy count.

**RNM-Gaussian Mechanism.** For a query $Q$ and classes 1 to $m$, let $o_j(Q) \in [m]$ denote the LLM prediction for $j$-th exemplar-query pair on $Q$, and $c_i(Q)$ denote the vote count for the $i$-th class, i.e., $c_i(Q) = |\{j : o_j(Q) = i\}|$. The Report-Noisy-Max with Gaussian noise (RNM-Gaussian) mechanism can be defined as: $\mathcal{M}_\sigma(Q) := \underset{j \in [m]}{\operatorname{argmax}} \{c_j(Q) + \mathcal{N}(0, \sigma^2)\}$ where $\mathcal{N}(0, \sigma^2)$ is the Gaussian distribution with mean 0 and variance $\sigma^2$. The aggregation mechanism selects the class with the highest vote count after adding Gaussian noise to each count. Intuitively, adding noise obfuscates the contribution of any single exemplar in a dataset. While there exist other mechanisms for RNM, in this work we adopt RNM-Gaussian due to the well-studied privacy cost analysis for the Gaussian mechanism. We use the state-of-the-art numerical privacy accountant (Gopi et al., 2021) for computing the overall privacy cost and we defer the details to Appendix B and Remark 1.

## 3.2 PRIVATE AGGREGATION FOR LANGUAGE GENERATION

Although our private voting method works well for text classification, extending it to the more compelling task of language *generation* proves to be non-trivial. In this section, we first describe the challenges of private language generation, namely the high-dimensional nature of the domain, and then describe our design goals to address these challenges. We then propose two novel techniques (Section 3.2.1 & Section 3.2.2) that we overview in Figure 4.

**Challenges of dimensionality in privately generating language.** An autoregressive language model generates text (conditioned on some prefix $\hat{x}_1, \ldots, \hat{x}_i$) by iteratively sampling $\hat{x}_{i+1} \sim \textbf{LLM}(x_{i+1}|\hat{x}_1, ..., \hat{x}_i)$ and then feeding $\hat{x}_{i+1}$ back into the model to sample $\hat{x}_{i+2} \sim \textbf{LLM}(x_{i+2}|\hat{x}_1, ..., \hat{x}_{i+1})$. This process is repeated until a desired stopping criterion is reached (e.g., the sentence length limit). The number of possible values that $\hat{x}_{i+1}$ can take on is equal to the vocabulary space of the model's tokenizer; consider a vocab size of $50,000$. The number of possible values that the *entire generation* can take on, for a maximum generation length of $100$, is therefore $50,000^{100}$, and this constitutes the size of the voting space for our private voting technique. It is unlikely that the model conditioned on two distinct exemplar pairs will generate the same text given a sufficiently long generation, because of the autoregressive nature of generation. Therefore, to assemble a histogram for language generation, where the "classes" are *all possible generations of a given length*, would yield an intractably large yet sparse histogram -precisely the opposite of what we want for our private voting method. The alternative is to operate the private voting method at each iteration, but this requires composing the privacy loss over the *number of tokens* being generated, which will quickly destroy the privacy-utility tradeoff.

**Design goal.** We fuse our insights into a design goal that will enable us to generate high-quality passages of text under privacy constraints. We want to do private aggregation in a lower dimensional

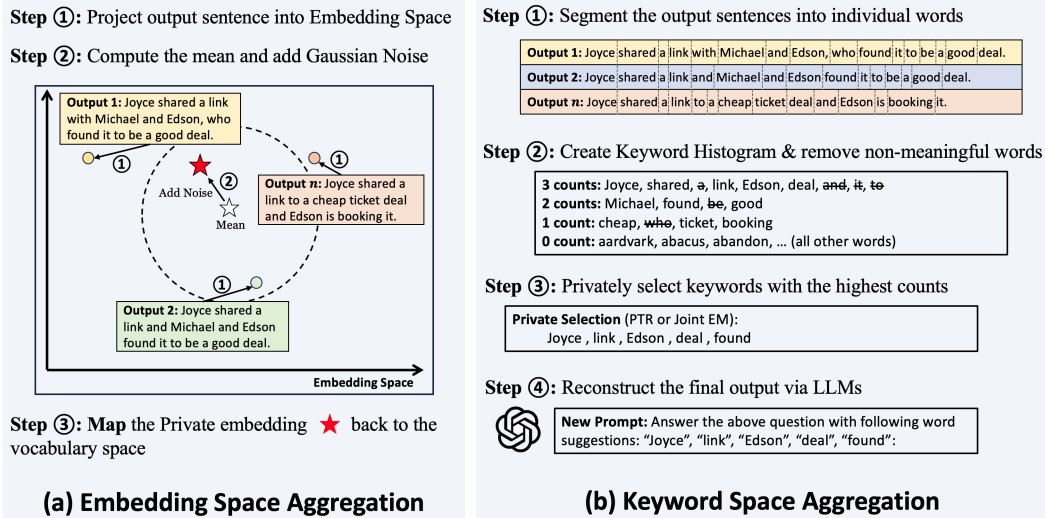

Figure 4: An overview of private aggregation methods for language generation. (a) **Embedding Space Aggregation** (Section 3.2.1): First, we transform all output sentences into an embedding space using a text-to-embedding model. Second, we privately estimate the mean embedding. Finally, we reconstruct a sentence from the privatized mean embedding. (b) **Keyword Space Aggregation** (Section 3.2.2): First, we decompose all output sentences into individual words and form a histogram based on their frequencies. Then, we employ either the PTR or joint EM mechanism to privately select the keywords. Finally, we reconstruct the sentence by incorporating these selected keywords into the prompt and re-querying the API.

space, by transforming generated model outputs into representations that preserve relative semantic meaning. We now propose a method that maps to the *embedding space* (Section 3.2.1) and a method that transforms model outputs into what we call the *keyword space* (Section 3.2.2).

### 3.2.1 EMBEDDING SPACE AGGREGATION (ESA)

Figure 4(a) depicts how embedding space aggregation (ESA) maps outputs to the embedding space and then reconstructs the private aggregation. The semantic embedding space (Reimers & Gurevych, 2019) is a natural choice for a representation that preserves the distance between outputs according to relative semantic meaning.

**Algorithm Overview.** We map each sentence generated by the LLM for a given exemplar-query pair onto the embedding space via a publicly available text-to-embedding model. In our empirical evaluations, we use OpenAI's widely used `text-embedding-ada-002` model[2], which maps each input sentence into a 1563-dimensional embedding vector with $\ell_2$ norm of 1. We then release a privatized mean of the embedding vectors converted from the generated sentences based on each private exemplar-query pair. To map from the embedding space back to a human-readable output space, we look for the sentences from the original sentence space that have similar embeddings to the newly privatized mean embedding.

**Technical Details.** Privately estimating the mean of the embedding vectors is straightforward; because we know the $\ell_2$ norm of all embeddings is 1, the *Gaussian mechanism* (Dwork et al., 2006a) minimizes the estimation error. The challenge lies in mapping the private mean from embedding space back to the sentence space so that we can output it to the user. Utilizing the LLM's zero-shot capabilities, we generate a set of sentence candidates by querying the API without any context. This approach ensures that these generated sentences do not add to the privacy budget, as they don't make use of the private in-context exemplars. We then select the generated sentence that maximizes the cosine similarity with the privatized mean embedding. The performance of our ESA technique depends on the quality of the methods used for the text-to-embedding and embedding-to-text mappings. While publicly available text-to-embedding models can generate good representations, going from *embedding-to-text* is an active research direction on its own (Morris et al., 2023; Linus,

---

[2]https://platform.openai.com/docs/guides/embeddings

2023). While there exist other mechanisms for RNM, in this work we adopt RNM-Gaussian due to the well-studied privacy cost analysis for the Gaussian mechanism. **Privacy analysis:** Since ESA also invokes a (subsampled) Gaussian mechanism, similar to text classification, we use the state-of-the-art numerical privacy accountant (Gopi et al., 2021) for computing the overall privacy cost and we defer the details to Appendix B and Remark 1.

### 3.2.2 KEYWORD SPACE AGGREGATION (KSA)

In this section, we introduce keyword space aggregation (KSA Figure 4(b)). KSA maps the model outputs into what we call the *keyword space*, performs private aggregation in the keyword space, and maps back to the output space by using the keywords to create a prompt for the LLM. The keyword space can be considered a low-dimensional approximation of the entire sentence space, and enables use of the private voting method without suffering from the curse of dimensionality.

**Algorithm Overview.** The goal of this algorithm is to extract a set of keywords that are very likely to be contained in a sentence that performs well as the answer for the query.[3] Clearly, such keywords should be present in many sentences generated based on different disjoint private in-context exemplars. Hence, we can count the frequency of each word token among the sentences generated based on individual private in-context exemplars and release the top-$K$ tokens that achieve the highest counts in a differentially private way. After obtaining those keywords, we can reconstruct a complete sentence by designing a new prompt with keywords and querying the LLM API.

**Technical Details.** Applying the RNM mechanism $K$ times to release the top-$K$ tokens based on count might seem straightforward. However, such an algorithm repeats RNM for $K$ times, and hence the privacy costs can be large for relatively large $K$. Moreover, it is very likely that the keyword space is large or even infinite. Fortunately, private top-$K$ selection on large domain spaces has been a well-studied problem, and we adopt two state-of-the-art methods for different scenarios depending on the size of the voting space. **(1) Moderately large domain space.** In this case, we adopt the joint exponential mechanism (joint EM) (Gillenwater et al., 2022). Unlike repeated applying RNM for $K$ times, this approach directly performs RNM on the space of all size-$K$ sequences and hence does not use composition. Note that for this case, the ranking of the counts for the word tokens is also released. **(2) Very large or infinite domain space.** In this case, we adopt the technique from Zhu & Wang (2022) which is based on the famous propose-test-release (PTR) paradigm. The main idea here is that, as long as the vote count difference between the $K$th and $(K+1)$th highest candidate is $> 2$, we can release the tokens with the top-$K$ vote counts directly without privatization. We note that in this case, the ranking of the counts for the word tokens is not released.

See Appendix A for a detailed description of our methods. **Privacy analysis:** For KSA, we are composing (Subsampled) Propose-Test-Release (PTR) paradigm and Exponential Mechanism (EM). Since the PRV is unknown for neither PTR paradigm nor EM, PRV accountant is not applicable here and we instead use the tool of (approximate) Renyi Differential Privacy (RDP) for calculating the final privacy guarantee. The detailed (approximate) RDP analysis for the PTR paradigm and Joint EM can be found in Appendix B.3. In particular, we derive the first subsampling amplification result for approximate RDP in Appendix B.3.3.

## 4 EXPERIMENTS

In this section, we demonstrate the experimental results of DP-ICL across three different tasks, including *text classification* (Section 4.1), *document question-answering* (Section 4.2), and *dialog summarization* (Section 4.3). Then, we perform ablation studies for *dialog summarization* in Section 4.4 and further results are presented in Appendix E & F.

### 4.1 DP-ICL FOR TEXT CLASSIFICATION

We study text classification using four datasets: sentiment analysis using **SST-2** (Socher et al., 2013) and **Amazon** (Zhang et al., 2015), topic classification using the 4-way **AGNews** (Zhang et al., 2015) datasets, and 6-way question classification using **TREC** (Voorhees & Tice, 2000). For all datasets, we randomly select 8,000 samples for training and 1,000 samples for testing if the size is large. We use the GPT-3 Babbage model for all tasks and additionally consider the GPT-3 Davinci model[4] for SST-2. We choose these models because they have shown promising results of in-context learning. Further details can be found in Appendix D.1.

---

[3]This idea is inspired by the Bag-of-Words method (Salton et al., 1975).

[4]GPT-3 Davinci has 100 times more parameters and is 40 times more expensive than GPT-3 Babbage.

We primarily focus on in-context learning with 4 exemplars (4-shot) and 10,000 queries. We compare with a zero-shot prediction that provides inherently privacy guarantee ($\varepsilon = 0$) and non-private ($\varepsilon = \infty$) 4-shot prediction. Since the performance of in-context learning has a large variance (Zhao et al., 2021; Min et al., 2022), we also compare our results with the performance of output aggregation ($\varepsilon = \infty$(Agg)). We set the number of exemplar-query pairs to 10 after subsampling and selected $\varepsilon = \{1, 3, 8\}$ and $\delta = 10^{-4}$ to achieve different levels of privacy.

**DP-ICL achieves a comparable performance with non-private ICL across all tasks (Table 2).** Our findings indicate that the impact of considering privacy on accuracy is marginal. For instance, the performance only drops by 0.04% for SST-2 with $\varepsilon = 3$ on GPT-3 Babbage. Even for a conservative privacy budget of $\varepsilon = 1$, we observe that DP-ICL can significantly outperform the zero-shot prediction (e.g., over 20 % for AGNews) depending on the dataset.

**DP-ICL can be further improved via deploying advanced LLMs.** By comparing the performance of GPT-3 Davinci and GPT-3 Babbage on SST-2, we find that the larger model leads to better performance across all $\varepsilon$ for DP-ICL. Take $\varepsilon = 1$ as an example; GPT-3 Davinci outperforms GPT-3 Babbage by $\sim$3.1%. In addition, we note that all our results can be further improved by simply replacing the GPT-3 API call with even more advanced LLMs, such as GPT-4.

Table 2: **Results of DP-ICL for Text classification.** We compare our method with zero-shot prediction ($\varepsilon = 0$), four-shot predictions ($\varepsilon = \infty$), and an aggregation of 10 four-shot predictions ($\varepsilon = \infty$ (Agg)). For $\varepsilon = \{1, 3, 8\}$, our DP-ICL generally surpasses zero-shot predictions and yields competitive performance relative to non-private predictions.

| Dataset | Model | $\varepsilon = 0$ (0-shot) | $\varepsilon = 1$ | $\varepsilon = 3$ | $\varepsilon = 8$ | $\varepsilon = \infty$ (Agg) | $\varepsilon = \infty$ |
|---|---|---|---|---|---|---|---|
| SST-2 | Babbage | 86.58 | $91.97_{0.49}$ | $92.83_{0.28}$ | $92.90_{0.24}$ | $92.87_{0.09}$ | $91.89_{1.23}$ |
| | Davinci | 94.15 | $95.11_{0.35}$ | $95.80_{0.21}$ | $95.83_{0.21}$ | $95.73_{0.13}$ | $95.49_{0.37}$ |
| Amazon | Babbage | 93.80 | $93.83_{0.33}$ | $94.10_{0.22}$ | $94.12_{0.20}$ | $94.10_{0.11}$ | $93.58_{0.64}$ |
| AGNews | Babbage | 52.60 | $75.49_{1.46}$ | $81.00_{1.14}$ | $81.86_{1.22}$ | $82.22_{2.16}$ | $68.77_{11.31}$ |
| TREC | Babbage | 23.00 | $24.48_{3.58}$ | $26.36_{5.19}$ | $26.26_{5.61}$ | $26.32_{5.33}$ | $27.00_{7.72}$ |

## 4.2 DP-ICL FOR DOCUMENT QUESTIONS ANSWERING

Then, we consider the document questions answering task, which aims to answer questions via reasoning a given document. We adopt a dataset that originates from a Privacy Preserving Federated Learning Document VQA (PFL-DocVQA) competition (Tito et al., 2023). We directly leverage the token extracted from the OCR model as the given context and use LLMs to generate answers to questions. Here, we use the open-source model OpenLLaMA-13B Geng & Liu (2023) and 1-shot ICL as a cost-effective choice to conduct experiments, and our methods are readily generalizable to other LLMs. We employ three metrics, ROUGE-1, BLEU, and normalized Levenshitein similarity, to comprehensively evaluate our proposed methods. Higher values in these metrics indicate better performance. See Appendix D.2 for more details and examples.

For baseline methods, we include evaluations for zero-shot prediction ($\varepsilon = 0$), 1-shot prediction ($\varepsilon = \infty$), and non-private aggregation ($\varepsilon = \infty$(Agg)) where we perform aggregation without noise. We compare embedding space aggregation and keyword space aggregation by PTR approaches.[5] The ensemble, query, and output candidate sizes are all set to 100, $\varepsilon = \{1, 3, 8\}$ and $\delta = 4 \times 10^{-6}$.

Table 3: **Results of DP-ICL Applied to Document Question Answering.** We use three baselines including zero-shot predictions ($\varepsilon = 0$), 1-shot ICL ($\varepsilon = \infty$), as well as non-private aggergation ($\varepsilon = \infty$ (Agg)). In the non-private aggregation setting, we use either embedding or keyword methods without adding privacy noise.

| Methods | Metrics | $\varepsilon = 0$ (0-shot) | $\varepsilon = 1$ | $\varepsilon = 3$ | $\varepsilon = 8$ | $\varepsilon = \infty$ (Agg) | $\varepsilon = \infty$ (1-shot) |
|---|---|---|---|---|---|---|---|
| Embedding | ROUGE-1 $\uparrow$ | 19.05 | $37.78_{0.35}$ | $37.91_{0.19}$ | $38.06_{0.15}$ | 37.97 | 50.68 |
| | BLEU $\uparrow$ | 4.42 | $6.49_{0.20}$ | $6.51_{0.04}$ | $6.54_{0.16}$ | 6.43 | 24.03 |
| | Levenshtein $\uparrow$ | 16.15 | $30.39_{0.50}$ | $30.71_{0.45}$ | $30.88_{0.06}$ | 30.94 | 49.30 |
| Keyword by PTR | ROUGE-1 $\uparrow$ | 19.05 | $59.92_{0.60}$ | $60.40_{0.50}$ | $60.66_{0.61}$ | 62.42 | 50.68 |
| | BLEU $\uparrow$ | 4.42 | $23.32_{0.51}$ | $23.67_{0.45}$ | $23.93_{0.45}$ | 25.10 | 24.03 |
| | Levenshtein $\uparrow$ | 16.15 | $51.47_{0.67}$ | $52.05_{1.06}$ | $52.47_{1.09}$ | 52.42 | 49.30 |

[5]Here, we did not implement the joint EM method, given that the output domain could potentially be infinite.

**DP-ICL achieves competitive results to non-private aggregations even with $\varepsilon$=1 (Table 3).** Remarkably, our empirical results suggest that adopting differential privacy does not lead to substantial performance degradation. For instance, the decline in ROUGE-1 scores when shifting from $\varepsilon = \infty$(Agg) to $\varepsilon = 1$ is less than 3% for keyword space aggregation (KSA) by PTR methods. For embedding space aggregation (ESA), the decrease is even more minimal at 0.19%.

Another noteworthy point is that the KSA method significantly surpasses the ESA approach, even exceeding the results of standard 1-shot ICL. This is because the consensus of keywords in output sentences leads to a more reliable answer. This performance drop in the ESA is mainly due to two factors: (1) information loss during projecting outputs into an embedding space, and (2) the lack of high-quality candidates generated by zero-shot predictions of OpenLLaMA-13B models. We think employing advanced LLMs and embedding reconstruction methods could mitigate these drawbacks.

## 4.3 DP-ICL FOR DIALOG SUMMARIZATION

We evaluate on the SAMSum dialog summarization dataset Gliwa et al. (2019). This task is much more challenging than previous tasks because the output can be multiple long sentences. We consider all three proposed methods: embedding space aggregation (ESA), keyword by PTR, and keyword by jointEM, using 4-shot ICL and GPT-3 Davinci API. For the keyword space aggregation (KSA), GPT-3 is again used to reconstruct the answers with extracted keywords within prompts. We compare three baselines: zero-shot learning, 4-shot ICL, and predictions of non-private aggregation. More details of the evaluation are in Appendix D.2.

Table 4: **Results of DP-ICL for dialog summarization.** We again compare with zero-shot predictions ($\varepsilon = 0$), 4-shot ICL ($\varepsilon = \infty$), as well as non-private aggregation ($\varepsilon = \infty$(Agg)). We report three variants of our private aggregation approaches with $\varepsilon = \{1, 3, 8\}$ and $\delta = 5 \times 10^{-5}$.

| Method | Metrics | $\varepsilon = 0$ (0-shot) | $\varepsilon = 1$ | $\varepsilon = 3$ | $\varepsilon = 8$ | $\varepsilon = \infty$ (Agg) | $\varepsilon = \infty$ (4-shot) |
|---|---|---|---|---|---|---|---|
| Embedding | ROUGE-1 $\uparrow$ | 35.31 | $38.21_{0.39}$ | $38.92_{0.24}$ | $39.62_{0.40}$ | 40.27 | 43.32 |
| | ROUGE-2 $\uparrow$ | 12.65 | $14.55_{0.66}$ | $15.18_{0.43}$ | $15.43_{0.46}$ | 16.52 | 19.08 |
| | ROUGE-L $\uparrow$ | 27.02 | $29.85_{0.61}$ | $30.86_{0.22}$ | $31.24_{0.45}$ | 32.29 | 34.78 |
| Keyword by joint EM | ROUGE-1 $\uparrow$ | 35.31 | $40.02_{0.37}$ | $40.98_{0.47}$ | $41.21_{0.58}$ | 42.40 | 43.32 |
| | ROUGE-2 $\uparrow$ | 12.65 | $15.67_{0.60}$ | $16.49_{0.79}$ | $16.31_{0.43}$ | 15.61 | 19.08 |
| | ROUGE-L $\uparrow$ | 27.02 | $30.46_{0.73}$ | $31.76_{0.26}$ | $31.84_{0.34}$ | 32.60 | 34.78 |
| Keyword by PTR | ROUGE-1 $\uparrow$ | 35.31 | $38.54_{0.47}$ | $39.09_{0.39}$ | $39.71_{0.21}$ | 41.03 | 43.32 |
| | ROUGE-2 $\uparrow$ | 12.65 | $14.42_{0.54}$ | $14.32_{0.45}$ | $14.60_{0.39}$ | 15.91 | 19.08 |
| | ROUGE-L $\uparrow$ | 27.02 | $29.58_{0.45}$ | $30.18_{0.44}$ | $30.56_{0.30}$ | 32.47 | 34.78 |

**Employing DP-ICL offers consistent advantages over zero-shot learning across all methods (Table 4).** Notably, under privacy constraints with $\varepsilon = 1$, our most effective approach, keyword by joint EM, yielded an improvement of approximately 4.5% in ROUGE-1, 3.0% in ROUGE-2, and 3.4% in ROUGE-L than zero-shot learning. This result is only marginally lower, by $\sim$1% on average than the performance achieved with non-private aggregations. Interestingly, the keyword by joint EM outperforms the keyword by PTR; this advantage is primarily due to joint EM also releasing the order of frequency. Another finding is that our non-private aggregation methods performed worse than in-context learning predictions, even for keyword space aggregation methods. We leave the question of optimizing the utilization of extracted keywords as future research.

## 4.4 ABLATION STUDIES

We also conduct an ablation study on the dialog summarization task, as it is the most challenging task, with varying number of queries and number of ensembles. Experimental results on text classification task are provided in Appendix E, and more findings related to language generation tasks are present in Appendix F. Here, we set the differential privacy parameter $\varepsilon = 3$.

**Effectiveness across numbers of ensembles (Figure 5(a)).** Our prior evaluation discussed in Section 4.3 utilizes an ensemble of 100 teachers for all methods. In this section, we vary the ensemble size from 10 to 100. It is noteworthy that increasing the ensemble size results in raised subsampling rates, thereby introducing additional noise into the aggregation process. At the same time, a larger ensemble size could also generate a more reliable consensus. We observe a clear trend of performance improvement when increasing ensembles for embedding space aggregation and keyword by PTR. However, the result of the keyword by joint EM approach shows more fluctuations and reaches a high performance with 30 ensembles.

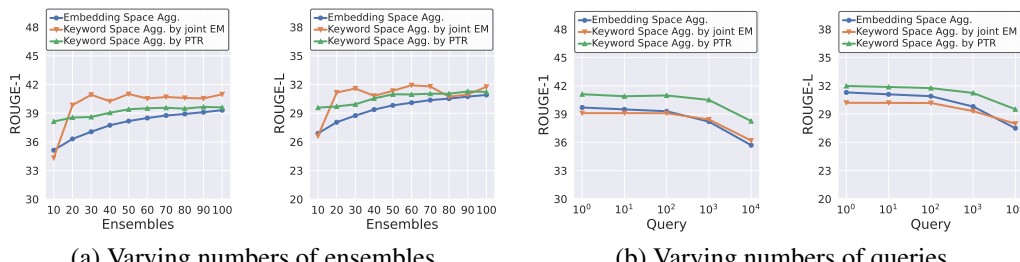

(a) Varying numbers of ensembles        (b) Varying numbers of queries

Figure 5: Ablation studies on dialog summarization task. Figure 13 & 12 present full results.

**Effectiveness across numbers of queries (Figure 5(b)).** We then investigate the impact of varying the number of queries across the set $\{1, 10, 100, 1000, 10000\}$. It is important to note that increasing the number of queries inherently strengthens privacy protection, causing a higher noise level. Our results indicate that performance degradation in KSA by joint EM remains marginal up to $10^3$ queries, suggesting its suitability for real-world applications.

## 5 RELATED WORKS

**Differentially Private Language Models.** The existing research on differentially private language models (Li et al., 2022; Yu et al., 2022; Bu et al., 2023; He et al., 2023) primarily focused on improving DP-SGD (Abadi et al., 2016) for training language models. In this paradigm, noise is introduced to the gradient during the model's training to ensure privacy. However, as the scale of the large language models significantly increased, fine-tuning has become much more challenging, making this approach less practical. We provide detailed qualitative and quantitative comparisons between DP-ICL and DP-SGD in Appendix C.

**Concurrent works on Differentially Private In-Context Learning.** For differentially private in-context learning, concurrent to our work, Duan et al. (2023a) propose an approach that privately labels a publicly available dataset and then uses the newly labeled data pairs as demonstrations, while our approach does not rely on public data. Later, Tang et al. (2023) present an approach that privately generates in-context exemplars directly via prompting and achieves effective ICL. It is essential to underscore that both approaches are **restricted to tasks involving limited label spaces**, such as text classification and word extraction. By contrast, we show that DP-ICL can obtain competitive performance on SAMSum and DocVQA, which are considerably more complex and challenging tasks in language generation. Our methodology and compelling results indicate that our methods can be broadly applied across various natural language processing tasks.

## 6 DISCUSSION AND FUTURE WORKS

In this paper, we initiate the study of incorporating in-context learning with differential privacy. We developed a unified framework for privatizing ICL based on the famous "sample-and-aggregate" paradigm, and we propose several instantiations for the private aggregation for the task of text classification and language generation.

While our method exhibits strong performance, there are multiple directions for future research. Specifically, for the Embedding Space Aggregation method, a more advanced embedding-to-text model may yield further improvements in the model performance. For instance, recent work (Morris et al., 2023) has shown promising results in reconstructing sentences directly from their embeddings. Additionally, DP-ICL relies on dividing the exemplars into disjoint subsets and queries the LLM with each of the subsets. As the number of subsets increases, the computational efficiency, while still significantly better compared with directly fine-tuning the LLM, will be larger. The efficiency-utility tradeoff for the choice of the number of subsets in DP-ICL is an interesting problem for future works. Furthermore, DP-ICL does not allow an infinite amount of queries. While we consider a substantial number of queries (up to 100,000) in our experiments, future investigations could aim to increase this capacity via tighter privacy accounting techniques for propose-test-release and Exponential mechanism.

ACKNOWLEDGMENTS

This work was supported in part by the National Science Foundation under grant CNS-2131938, the Princeton SEAS Innovation Grant, and Princeton Gordon Y. S. Wu Fellowship.

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

# A   DETAILS AND PSEUDOCODE OF DP IN-CONTEXT LEARNING IN SECTION 3

In this section, we provide full details of our DP-ICL algorithms. These algorithms include text classification through RNM-Gaussian (Appendix A.1), language generation via embedding space aggregation (Appendix A.2), and language generation via keyword space aggregation (Appendix A.3).

---

**Algorithm 1** Differentially Private In-Context Learning (Meta Algorithm)

---

**Require:** Full private dataset $\mathcal{D}$, query set $\mathcal{Q} = \{Q\}$, model **LLM**, task description task, technique technique.

1: **for** $Q \in \mathcal{Q}$ **do**
2:     Subsample $D \leftarrow \mathcal{D}$.
3:     **if** task="text classification" **then**
4:         **Execute** RNM-Gaussian (Algorithm 2) on $D$ and $Q$.
5:     **else if** task="language generation", technique="ESA" **then**
6:         **Execute** ESA (Algorithm 3) on $D$ and $Q$.
7:     **else if** task="language generation", technique="KSA" **then**
8:         **Execute** KSA (Algorithm 4) on $D$ and $Q$.
9:     **end if**
10: **end for**

---

## A.1   TEXT CLASSIFICATION VIA RNM-GAUSSIAN

Our method is detailed in Algorithm 2, and further privacy analysis is detailed in Appendix B.1.

---

**Algorithm 2** RNM-Gaussian

---

**Require:** Private data $D$, query $Q$, model **LLM**, noise $\sigma$, number of subsets $N$

1: **Partition** $D_1, D_2, \ldots, D_N \leftarrow D$.
2: **for** $i \in \{1, \ldots, N\}$ **do**
3:     Form exemplar-query pair $D_i^Q = D_i \cup \{Q\}$.
4:     Obtain model output $O_i(Q) = \mathbf{LLM}(D_i^Q)$.
5:     Convert $O_i(Q)$ to a one-hot vector with length equal to the number of classes.
6: **end for**
7: Sum the one-hot vectors into a histogram $\mathbf{H}$.
8: Add noise to $\mathcal{N}\left(0, \sigma^2\right)$ to each entry of $\mathbf{H}$.
9: Report the top-1 bin from $\mathbf{H}$.

---

## A.2   LANGUAGE GENERATION VIA EMBEDDING SPACE AGGREGATION (ESA)

In Algorithm 3, we present the full descriptions of our ESA method. The main idea is to project those output sentences into embedding space, get a differentially private mean, and map it back to sentence space. Further privacy analysis, including how to compute $\sigma$, is presented in Appendix B.2.

## A.3   LANGUAGE GENERATION VIA KEYWORD SPACE AGGREGATION (KSA)

The keyword space aggregation (KSA) algorithm is demonstrated in Algorithm 4. We illustrate two differential private approaches for selecting keywords in the following subsections, including joint Exponential Mechanism (Appendix A.3.1) and Propose-Test-Release (Appendix A.3.2).

### A.3.1   KSA VIA JOINT EXPONENTIAL MECHANISM.

The main idea of the joint exponential mechanism (Gillenwater et al., 2022) is to provide a mechanism that samples *sequences* of items rather than use variants of the exponential mechanism that may require composition over the number of tokens (Durfee & Rogers, 2019). We create the "public domain" for our summarization tasks by creating a histogram of counts for the words in the dialogue we want to summarize. We increase these counts for each exemplar. Note that just creating a

---

**Algorithm 3** Embedding Space Aggregation

---

**Require:** Private data $D$, query $Q$, model **LLM**, noise $\sigma$, number of subsets $N$, public candidate sentences obtained by zero-shot predictions $O_C$ with total number of $C$

1: **Partition** $D_1, D_2, \ldots, D_N \leftarrow D$.
2: **for** $i \in \{1, \ldots, N\}$ **do**
3:     Form exemplar-query pair $D_i^Q = D_i \cup \{Q\}$.
4:     Obtain model output sentence $O_i(Q) = \textbf{LLM}(D_i^Q)$.
5:     Project $O_i(Q)$ into embedding vector $E_i(Q)$.
6: **end for**
7: Take the mean of all embedding vectors $E = \frac{1}{N} \sum_{i=1}^{N} E_i$.
8: Adding noise and obtain the noisy embedding $\widehat{E} = E + \mathcal{N}\left(0, \sigma^2 I\right)$.
9: **Return** the sentence $\text{argmax}_{o \in O_C}$ **CosineSimilarity**$(o, \widehat{E})$.

---

**Algorithm 4** Keyword Space Aggregation

---

**Require:** Private data $D$, query $Q$, model **LLM**, noise $\sigma$, number of subsets $N$, public candidates obtained by zero-shot predictions $O_C$ with total number of $C$, maximum token length $M$, method $\in \{\text{JEM}, \text{PTR}\}$.

1: **Partition** $D_1, D_2, \ldots, D_N \leftarrow D$.
2: **for** $i \in \{1, \ldots, N\}$ **do**
3:     Form exemplar-query pair $D_i^Q = D_i \cup \{Q\}$.
4:     Obtain model output sentence $O_i(Q) = \textbf{LLM}(D_i^Q)$.
5: **end for**
6: For each token, count the number of sentences in $\{O_i(Q)\}$ it appears, and form a histogram **H**.
7: **if** method = JEM **then**
8:     **Return** JointEM$(k, \textbf{H})$. (Algorithm 5)
9: **else if** method = PTR **then**
10:     $\widehat{k} = $ FindBestK$(\textbf{H})$. (Algorithm 7)
11:     **Return** TopKwithPTR$(\widehat{k}, $ FindBestK$(\textbf{H}))$. (Algorithm 6)
12: **end if**

---

**Algorithm 5** JointEM (Gillenwater et al., 2022)

**Require:** Vector of item counts $c_1, \ldots, c_d$, number of items to estimate $k$, privacy parameter $\varepsilon$

1:  Sort and relabel items so $c_1 \geq c_2 \geq \cdots \geq c_d$
2:  Construct matrix $\tilde{U}$ by $\tilde{U}_{ij} = -(c_i - c_j) - \frac{d(k-i)+j}{2dk}$
3:  Sort $\tilde{U}$ in decreasing order to get $\tilde{U}_{(1)}, \ldots, \tilde{U}_{(dk)}$, storing the (row, column) of each $\tilde{U}_{(a)}$ as $(r(a), c(a))$
4:  Initialize $n_1, \ldots, n_k \leftarrow 0$
5:  Initialize set of non-zero $n_i$, $N \leftarrow \emptyset$
6:  Initialize $b \leftarrow 0$
7:  **for** $a = 1, \ldots, dk$ **do**
8:      $n_{r(a)} \leftarrow c(a) - (r(a) - 1)$
9:      $N \leftarrow N \cup \{r(a)\}$
10:     **if** $|N| = k$ **then**
11:         break
12:     **end if**
13:     Set $\tilde{m}(\tilde{U}_{(a)}) \leftarrow 0$, and set $b \leftarrow a$
14: **end for**
15: Set $p \leftarrow \prod_{r \in [k]} n_r$
16: Compute $\tilde{m}(\tilde{U}_{(b+1)}) \leftarrow p / n_{r(a)}$
17: **for** $a = b + 2, \ldots, dk$ **do**
18:     Set $p \leftarrow p / n_{r(a)}$
19:     Compute $\tilde{m}(\tilde{U}_{(a)}) \leftarrow p$
20:     Update $n_{r(a)} \leftarrow n_{r(a)} + 1$
21:     Update $p \leftarrow p \cdot n_{r(a)}$
22: **end for**
23: Sample a utility $\tilde{U}_{ij}$ from: $\mathbb{P}\left[\tilde{U}_{ij}\right] \propto \tilde{m}\left(\tilde{U}_{ij}\right) \exp\left(\frac{\varepsilon\left[\tilde{U}_{ij}\right]}{2}\right)$
24: Initialize size-$k$ output vector $s$ with $s_i \leftarrow j$
25: **for** $i' = 1, 2, \ldots, i - 1, i + 1, \ldots, k$ **do**
26:     Compute $t_{i'}(\tilde{U}_{ij})$ by iterating through row $i'$ of $\tilde{U}$
27:     Sample $s_{i'}$ uniformly from $t_{i'}(\tilde{U}_{ij}) \setminus \{j, s_1, s_2, \ldots, s_{i'-1}\}$
28: **end for**
29: Return Vector of item indices $s$

histogram over the outputs of all exemplars would violate privacy. This is a challenge in extending KSA-JOINT to the infinite domain. We use this histogram to initialize a data structure to efficiently sample with the joint exponential mechanism. Our implementation uses the code from (Gillenwater et al., 2022).

### A.3.2 KSA VIA PROPOSE-TEST-RELEASE (KSA-PTR)

**Notations.** We use $N$ to denote the total number of tokens (e.g., $N = 50,000$). We use $\mathbf{H}$ to denote the histogram for the counts of each token, and we use $\mathbf{H}_{(j)}$ to denote the $j$th highest count, i.e., $\mathbf{H}_{(1)} \geq \mathbf{H}_{(2)} \geq \ldots \geq \mathbf{H}_{(N)}$.

The main idea of KSA-PTR is that, for the task of releasing the top-$k$ index set of a voting histogram, if $\mathbf{H}_{(k)} - \mathbf{H}_{(k+1)} > 2$, then the top-$k$ indices are exactly the same for all the neighboring datasets. Hence, one can release the exact top-$k$ indices without any randomness. However, we need to test whether $\mathbf{H}_{(k)} - \mathbf{H}_{(k+1)} > 2$ in a differentially private way, where we can leverage the famous propose-test-release paradigm (Dwork & Lei, 2009), as shown in Algorithm 6.

---

**Algorithm 6** TopKwithPTR

---

**Require:** $k$ – the number of top counted tokens to release; $\mathbf{H}$ – histogram for the counts of each token; $\delta$ – failure probability
 1: **Set** $d_k := \mathbf{H}_{(k)} - \mathbf{H}_{(k+1)}$.
 2: **Set** $\widehat{d}_k := \max(2, d_k) + \mathcal{N}(0, 4\sigma^2) - \Phi(1 - \delta; 0, 2\sigma)$.
 3: **If** $\widehat{d}_k > 2$, **Return** the exact top-$k$ tokens.
 4: **Else** Terminate (or use zero-shot learning).

---

As we can see, such an algorithm can have the highest utility when we choose the $k$ that maximizes $\mathbf{H}_{(k)} - \mathbf{H}_{(k+1)}$. Hence, to further improve the utility of the algorithm, we can select $k$ in a data-dependent way, i.e., we release $\operatorname{argmax}_k \mathbf{H}_{(k)} - \mathbf{H}_{(k+1)}$ in a differentially private way (which is another Report-Noisy-Max) using Exponential mechanism.

---

**Algorithm 7** FindBestK

---

**Require:** $\mathbf{H}$ – histogram for the counts of each token
 1: Compute histogram gap $d_k := \mathbf{H}_{(k)} - \mathbf{H}_{(k+1)}$ for each $k = 1 \ldots N - 1$.
 2: **Return** $\operatorname{argmax}_k \{d_k + r(k) + \texttt{Gumbel}(4/\varepsilon)\}$

---

Here, $r(k)$ is a regularizer independent of the dataset, e.g., we can set $r(k) = -\infty$ for any $k > 30$ and $k < 15$, if we don't want to return more than 30 or less than 15 tokens.

# B  Privacy Analysis

In this section, we review the important properties of differential privacy and provide the privacy analysis for the algorithms introduced in Section 3.

**Outline of this section:** In this section, we first review the concept of DP as well as the necessary background of DP composition. The privacy analysis for differentially private text classification (RNM-Gaussian) is provided in Appendix B.1. The privacy analysis for ESA is provided in Appendix B.2. The privacy analysis for KSA-Joint EM and KSA-PTR is provided in Appendix B.3.1 and B.3.2, respectively.

We first state the formal DP definition.

**Definition 1** (Differential Privacy (Dwork et al., 2006b)). *For $\varepsilon, \delta \geq 0$, a randomized algorithm $\mathcal{M} : \text{MultiSets}(\mathcal{X}) \rightarrow \mathcal{Y}$ is $(\varepsilon, \delta)$-differentially private if for every neighboring dataset pair $D, D' \in \text{MultiSets}(\mathcal{X})$, we have:*

$$\forall\, T \subseteq \mathcal{Y} \ \Pr[\mathcal{M}(D) \in T] \leq e^{\varepsilon} \cdot \Pr[\mathcal{M}(D') \in T] + \delta$$

*where the randomness is over the coin flips of $\mathcal{M}$.*

**Post-processing Property.** Differential privacy exhibits a robust post-processing property. Informally, this means that if a mechanism is differentially private, then any post-processing applied to the output of that mechanism is also differentially private. This property is crucial for enabling flexible analysis of privately released data.

**Lemma 2** (Post-processing (Dwork et al., 2006b)). *If $\mathcal{M} : \text{MultiSets}(\mathcal{X}) \rightarrow \mathcal{Y}$ is $(\varepsilon, \delta)$-differentially private and $f : \mathcal{Y} \rightarrow \mathcal{Z}$ is an arbitrary (possibly randomized) function, then the composed mechanism $f \circ \mathcal{M} : \text{MultiSets}(\mathcal{X}) \rightarrow \mathcal{Z}$ is also $(\varepsilon, \delta)$-differentially private.*

**(Adaptive) Composition of Differential Privacy.** In practice, multiple differentially private mechanisms may be applied to the same dataset. Crucially, multiple DP mechanisms can be *adaptively* composed in the sense that the output of one mechanism can be used as an input to another mechanism, denoted as $\mathcal{M}(D) = \mathcal{M}_1 \circ \mathcal{M}_2(D) := (\mathcal{M}_1(D), \mathcal{M}_2(D, \mathcal{M}_1(D)))$. Differential privacy offers strong composition guarantees, that help quantify the cumulative privacy loss resulting from these combined mechanisms. These guarantees are provided by various composition theorems or privacy accounting techniques, including the basic composition theorem (Dwork et al., 2006a), advanced composition theorem (Dwork et al., 2010), and Moments Accountant (Abadi et al., 2016). For example, the basic composition theorem states that if $\mathcal{M}_1$ is $(\varepsilon_1, \delta_1)$-DP and $\mathcal{M}_2$ is $(\varepsilon_2, \delta_2)$-DP, then the adaptive composition of $\mathcal{M}_1$ and $\mathcal{M}_2$ is $(\varepsilon_1 + \varepsilon_2, \delta_1 + \delta_2)$-DP.

Consider two attackers: the first asks their allotted $k$ queries in one batch and then observes the answers, the second asks each query sequentially and incorporates information gained from observing the answer to the current query into the next query. The second attacker is certainly stronger, and this increased strength is captured by adaptive composition.

**Privacy Amplification by Subsampling.** Privacy amplification by subsampling is a technique used to enhance privacy guarantees in differentially private mechanisms by randomly selecting a subset of the data before applying the privacy mechanism. This subsampling process can lead to a reduction in the privacy cost, allowing for better utility while preserving privacy. We can show that the Poisson subsampled Gaussian mechanism with sensitivity 1, noise scale $\sigma$, and subsampling rate $q$ has the PRV $Y = \log(P(o)/Q(o)), o \sim P$, where $P = (1 - q)\mathcal{N}(0, \sigma^2) + q\mathcal{N}(1, \sigma^2)$ and $Q = \mathcal{N}(0, \sigma^2)$, and $P(\cdot), Q(\cdot)$ are the density functions of $P, Q$. With the PRV of subsampled Gaussian mechanism as well as the PRV accountant, we can now efficiently and tightly track the privacy costs for DP-ICL.

## B.1  text classification via RNM-Gaussian

**Theorem 3.** *The mechanism RNM-Gaussian $\mathcal{M}_\sigma$ from Section 3.1 is $(\varepsilon, \delta)$-DP with $\sigma = 2\sqrt{\log(1.25/\delta)}/\varepsilon$.*

*Proof.* Note that $\mathcal{M}_\sigma$ can be broken down into applying the $\text{argmax}$ operator on a noisy histogram, which is generated by adding Gaussian noise to each dimension of the original histogram. The Gaussian mechanism is known to satisfy $(\varepsilon, \delta)$-DP with $\sigma = \Delta\sqrt{2\log(1.25/\delta)}/\varepsilon$ (Dwork et al.,

2014), where $\Delta := \sup_{D \sim D'} \|f(D) - f(D')\|$ represents the global sensitivity of the underlying aggregation function $f$. In our case, $f$ calculates the original voting histogram. As each exemplar-query prediction may alter two counts (increasing one and decreasing the other), the sensitivity $\Delta$ is $\sqrt{2}$. The overall privacy guarantee is then derived from the post-processing property of differential privacy. $\qquad\square$

## B.2 Embedding Space Aggregation (ESA)

**Theorem 4.** *The Step 2 to Step 9 in Alg. 3 is $(\varepsilon, \delta)$-DP with $\sigma = 2\sqrt{\log(1.25/\delta)}/\varepsilon$.*

*Proof.* Note that each embedding output by the text-to-embedding model has $\ell_2$ norm to be 1. Hence, the referred steps are essentially the same as Gaussian mechanism with $\ell_2$ sensitivity 1. The last step of releasing the public candidate sentence that has the maximum cosine similarity can be regarded as the post-processing step and hence does not affect the overall privacy guarantee. $\qquad\square$

**Remark 1** (**Tracking Privacy Loss with PRV Accountant for Subsampled Gaussian mechanism**). *To better keep track of the privacy cost for RNM-Gaussian and ESA, we use the most recent advances in privacy cost accounting based on the notion of the Privacy Loss Random Variable (PRV) (Dwork & Rothblum, 2016). The PRV accountant was introduced by Koskela et al. (2020) and later refined in Koskela & Honkela (2021); Gopi et al. (2021). For any DP-algorithm, one can easily compute its $(\varepsilon, \delta)$ privacy guarantee based on the distribution of its PRV. The key property of PRVs is that, under (adaptive) composition, they simply add up; the PRV $Y$ of the composition $\mathcal{M} = \mathcal{M}_1 \circ \mathcal{M}_2 \circ \cdots \circ M_k$ is given by $Y = \sum_{i=1}^k Y_i$, where $Y_i$ is the PRV of $\mathcal{M}_i$. Therefore, one can then find the distribution of $Y$ by convolving the distributions of $Y_1, Y_2, \ldots, Y_k$. Prior works (Koskela & Honkela, 2021; Gopi et al., 2021) approximate the distribution of PRVs by truncating and discretizing them, then using the Fast Fourier Transform (FFT) to efficiently convolve the distributions.*

## B.3 Language Generation via Keyword Space Aggregation (KSA)

Rényi differential privacy (RDP) is a variant of the standard $(\varepsilon, \delta)$-DP that uses Rényi-divergence as a distance metric between the output distributions of $\mathcal{M}(D)$ and $\mathcal{M}(D')$, which is particularly useful in training differentially private machine learning models.

**Definition 5** (Rényi Differential Privacy (Mironov, 2017))**.** *We say that a mechanism $\mathcal{M}$ is $(\alpha, \varepsilon_{\mathcal{M}}(\alpha))$-RDP with order $\alpha \in (1, \infty)$ if for every dataset pair $D, D' \in \mathrm{MultiSets}(\mathcal{X})$ such that $d(D, D') = 1$, we have:*

$$D_\alpha\left(\mathcal{M}(D) \| \mathcal{M}(D')\right) := \frac{1}{\alpha - 1} \log \mathbb{E}_{o \sim \mathcal{M}(D')}\left[\left(\frac{\mu_{\mathcal{M}(D)}(o)}{\mu_{\mathcal{M}(D')}(o)}\right)^\alpha\right] \le \varepsilon_{\mathcal{M}}(\alpha) \qquad (1)$$

*where $\mu_{\mathcal{M}}(\cdot)$ denotes the density function of $\mathcal{M}$'s distribution.*

Another useful relaxation of the RDP definition is approximate RDP.

**Definition 6** (Approximate RDP (Bun & Steinke, 2016; Zhu & Wang, 2022))**.** *We say a randomized algorithm $\mathcal{M}$ is $\delta$-approximately $(\alpha, \varepsilon_{\mathcal{M}}(\alpha))$-RDP with order $\alpha \ge 1$, if for all neighboring dataset $D, D'$, there exist events $E$ (depending on $\mathcal{M}(D)$) and $E'$ (depending on $\mathcal{M}(D')$) such that $\Pr[E] \ge 1 - \delta$ and $\Pr[E'] \ge 1 - \delta$, and $\forall \alpha \ge 1$, we have*

$$D_\alpha\left(\mathcal{M}(D) | E \| \mathcal{M}(D') | E'\right) \le \varepsilon_{\mathcal{M}}(\alpha) \qquad (2)$$

For both methods of KSA-JEM and KSA-PTR, we use RDP and approximate RDP for a tighter measure of the privacy cost under composition. After we obtain the (approximate) RDP guarantee for the overall algorithm, we can then convert the privacy guarantee back into the standard DP definition. We refer the readers to Bun & Steinke (2016) and Mironov (2017) for the composition and conversion formula for RDP and approximate RDP. In the following, we state the privacy guarantee of individual building blocks for private prompt generation and selection in terms of (approximate) RDP.

We then introduce the exponential mechanism (McSherry & Talwar, 2007), one of the most famous and frequently used DP mechanisms. The exponential mechanism takes a utility function $q : \mathrm{MultiSets} \times \mathcal{Y} \to \mathbb{R}$ and can be thought of as evaluating how good $q(D, y)$ is for an outcome $y \in \mathcal{Y}$ on dataset $D$.

**Definition 7** (Exponential Mechanism). *Let* $\mathsf{EM}_q : \text{MultiSets} \to \mathcal{Y}$ *be a mechanism where for all outputs* $y \in \mathcal{Y}$ *we have*

$$\Pr[\mathsf{EM}_q(D) = y] \propto \exp\left(\frac{\varepsilon}{2\Delta(q)} q(D, y)\right)$$

*where* $\Delta(q)$ *is the sensitivity of the quality score, i.e. for all neighboring inputs* $D, D'$ *we have* $\sup_{y \in \mathcal{Y}} |q(D, y) - q(D', y)| \le \Delta(q)$

Furthermore, Durfee & Rogers (2019) shows that adding Gumbel noise to each output's utility and releasing the output with the highest noisy utility score is equivalent to using the exponential mechanism.

**Theorem 8** (Bun & Steinke (2016)). *The exponential mechanism is* $\varepsilon$-DP, and $(\alpha, \varepsilon_{\mathsf{EM}}(\alpha))$-RDP s.t.

$$\varepsilon_{\mathsf{EM}}(\alpha) := \min\left(\frac{\alpha}{2}\varepsilon^2, \frac{1}{\alpha - 1} \log\left(\frac{\sinh(\alpha\varepsilon) - \sinh((\alpha - 1)\varepsilon)}{\sinh(\varepsilon)}\right)\right)$$

### B.3.1 KSA via Joint Exponential Mechanism (KSA-Joint EM)

**Theorem 9.** *Alg. 5 is* $\varepsilon$-DP, and $\varepsilon_{EM}(\alpha)$-RDP.

*Proof.* Alg. 5 is an Exponential mechanism on the domain space of positive integers $k = 1, 2, \ldots$, where the utility of $k$ is $c_k$. The sensitivity of $d_k$ is 1 since each exemplar-query prediction may alter a count $c_k$ at most 1. Hence, the DP and RDP guarantee follows from the privacy guarantee of exponential mechanism in Theorem 8. □

### B.3.2 KSA via Propose-Test-Release (KSA-PTR)

**Theorem 10.** *Alg. 7 is* $\varepsilon$-DP, and $\varepsilon_{EM}(\alpha)$-RDP.

*Proof.* Alg. 7 is an Exponential mechanism on the domain space of positive integers $k = 1, 2, \ldots$, where the utility of $k$ is $d_k := \mathbf{H}_{(k)} - \mathbf{H}_{(k+1)}$. The sensitivity of $d_k$ is 2. Hence, the DP and RDP guarantee follows from the privacy guarantee of exponential mechanism in Theorem 8. □

**Theorem 11.** *Alg. 6 is* $\delta$-approximate $\frac{\alpha}{2\sigma^2}$-RDP.

*Proof.* Releasing the noisy threshold $\widehat{d_k}$ is $\frac{\alpha}{2\sigma^2}$-RDP.

If $d_k > 2$, then releasing the exact top-$k$ tokens has no privacy cost, as its local sensitivity is 0.

If $d_k \le 2$, then if $\widehat{d_k} \le 2$, the program terminates and there's no privacy cost.

If $d_k \le 2$, the failure probability

$$
\begin{aligned}
\Pr[\widehat{d_k} > 2] &= \Pr[\max(2, d_k) + \mathcal{N}(0, 4\sigma^2) - \Phi(1 - \delta; 0, 2\sigma) > 2] \\
&= \Pr[2 + \mathcal{N}(0, 4\sigma^2) - \Phi(1 - \delta; 0, 2\sigma) > 2] \\
&= \Pr[\mathcal{N}(0, 4\sigma^2) - \Phi(1 - \delta; 0, 2\sigma) > 0] \\
&= \delta
\end{aligned}
$$

□

### B.3.3 Privacy amplification by subsampling for approximate RDP

In the following, we present the privacy amplification of approximate RDP by Poisson subsampling. To the best of our knowledge, we are the first to derive the following result.

**Theorem 12.** *If* $\mathcal{M}$ *is* $\delta$-approximate $\varepsilon_{\mathcal{M}}(\alpha)$-RDP, then $\mathcal{M} \circ$ Poisson *with subsampling rate* $q$ *is* $\delta q$-approximate $\varepsilon_{\mathcal{M}\circ\text{Poisson}}(\alpha)$-RDP, where $\varepsilon_{\mathcal{M}\circ\text{Poisson}}(\alpha)$ *is the tightest possible amplification bound for any mechanism that is* $\varepsilon_{\mathcal{M}}(\alpha)$-RDP *with subsampling rate* $\frac{q(1-\delta)}{1-q\delta}$.

*Proof.* Consider $D := D' \cup \{z\}$, $D, D'$ are neighboring datasets. Denote $S \subseteq D'$, and let $\gamma_S$ the probability of sampling $S$. Denote $\mu_S := \mathcal{M}(S)$.

$$\mathcal{M}(\text{Poisson}(D')) = \sum_{S \subseteq D'} \gamma_S \mu_S \tag{3}$$

$$\mathcal{M}(\text{Poisson}(D)) = \sum_{S \subseteq D'} \gamma_S \left((1-q)\mu_S + q\mu_{S \cup \{z\}}\right) \tag{4}$$

$$= (1-q) \sum_{S \subseteq D'} \gamma_S \mu_S + q \sum_{S \subseteq D'} \gamma_S \mu_{S \cup \{z\}} \tag{5}$$

By definition of approximate RDP, for any pair of $S, S \cup \{z\}$, we have event $E_S, E_{S \cup \{z\}}$ s.t. $D_\alpha(\mu_S | E_S \| \mu_{S \cup \{z\}} | E_{S \cup \{z\}}) \leq \varepsilon_{\mathcal{M}}(\alpha)$ and $\Pr[E_S] = 1 - \delta$ and $\Pr[E_{S \cup \{z\}}] = 1 - \delta$. Hence, we can rewrite $\mathcal{M}(\text{Poisson}(D'))$ and $\mathcal{M}(\text{Poisson}(D))$ as

$$\mathcal{M}(\text{Poisson}(D')) = (1-q) \sum_{S \subseteq D'} \gamma_S \mu_S + q \sum_{S \subseteq D'} \gamma_S \mu_S$$

$$= (1-q) \sum_{S \subseteq D'} \gamma_S \mu_S + q \sum_{S \subseteq D'} \gamma_S \left((1-\delta)\mu_S | E_S + \delta\mu_S | \bar{E}_S\right)$$

$$= (1-q) \sum_{S \subseteq D'} \gamma_S \mu_S + q(1-\delta) \sum_{S \subseteq D'} \gamma_S \mu_S | E_S + q\delta \sum_{S \subseteq D'} \gamma_S \mu_S | \bar{E}_S$$

$$\mathcal{M}(\text{Poisson}(D)) = (1-q) \sum_{S \subseteq D'} \gamma_S \mu_S + q \sum_{S \subseteq D'} \gamma_S \mu_{S \cup \{z\}}$$

$$= (1-q) \sum_{S \subseteq D'} \gamma_S \mu_S + q(1-\delta) \sum_{S \subseteq D'} \gamma_S \mu_{S \cup \{z\}} | E_{S \cup \{z\}} + q\delta \sum_{S \subseteq D'} \gamma_S \mu_{S \cup \{z\}} | \bar{E}_{S \cup \{z\}}$$

Hence, there exists event $E_D, E_{D'}$ s.t. $\Pr[E_D] \geq 1 - q\delta$ and $\Pr[E_{D'}] \geq 1 - q\delta$, and

$$\mathcal{M}(\text{Poisson}(D')) | E_D = \frac{(1-q)}{1-q\delta} \sum_{S \subseteq D'} \gamma_S \mu_S + \frac{q(1-\delta)}{1-q\delta} \sum_{S \subseteq D'} \gamma_S \mu_S | E_S$$

$$\mathcal{M}(\text{Poisson}(D)) | E_{D'} = \frac{(1-q)}{1-q\delta} \sum_{S \subseteq D'} \gamma_S \mu_S + \frac{q(1-\delta)}{1-q\delta} \sum_{S \subseteq D'} \gamma_S \mu_{S \cup \{z\}} | E_{S \cup \{z\}}$$

Hence

$$D_\alpha \left(\mathcal{M}(\text{Poisson}(D)) | E_D \| \mathcal{M}(\text{Poisson}(D')) | E_{D'}\right) \tag{6}$$

has privacy amplification with subsampling rate $\frac{q(1-\delta)}{1-q\delta}$. $\qquad \square$

## C   DP-ICL ENABLES PRIVATE PREDICTION

Our work represents a major departure from prior work on DP LLMs in that we consider private *prediction* rather than private training. A line of recent work (Li et al., 2022; Yu et al., 2022; Bu et al., 2022; He et al., 2023) has proposed fine-tuning pre-trained models on downstream tasks with differentially private stochastic gradient descent (DP-SGD) (Abadi et al., 2016). Despite ample research into DP LLMs and the growing industry demand for solutions to augment LLMs with proprietary data (Kuchaiev et al., 2019; Nvidia, 2023), a number of key challenges remain for DP LLMs that we seek to address by considering *private prediction*.

**Private training makes training harder.** Fine-tuning with DP-SGD requires adopting entirely new hyperparameters and shifting existing hyperparameters to be radically different from non-private training (Li et al., 2022). Performing this additional hyperparameter tuning can take hundreds of trials. DP-SGD uses per-example gradient clipping to bound the sensitivity of individual datapoints. Materializing per-example gradients can increase the memory consumption of training by an order of magnitude (Bu et al., 2022) and slow down training. Although recent methods have been proposed for efficient hyperparameter tuning (Panda et al., 2022; Papernot & Steinke, 2022), efficient per-example gradient clipping (Li et al., 2022), and parameter-efficient fine-tuning (Yu et al., 2022), we emphasize that DP-SGD introduces challenging engineering and optimization problems that are a topic of ongoing research. **Our method requires no hyperparameter tuning and is computationally efficient.**

**Private training is incompatible with black-box LLMs.** Developers building on top of cloud-hosted LLMs such as OpenAI, Anthropic, or AWS Bedrock cannot implement the complex DP-SGD algorithm (Sivasubramanian, 2023). Organizations employing closed-source LLMs such as GPT-3+, Claude, or Bard cannot even access the weights for fine-tuning and may never be able to (OpenAI, 2023). **Our method is compatible with any LLM API.**

**Private training does not allow flexible data editing.** Private training generates a single model that is inextricably tied to each datapoint in its training data. This is at odds with the right to be forgotten mandated by GDPR (Politico, 2023), that would require retraining the entire model to delete the influence of a private datapoint -an impracticality if not an outright impossibility when considering fine-tuning billion-parameter models. By contrast, honoring the right to be forgotten with DP-ICL is as straightforward as just removing the individual's private data from the exemplar database. **Our method enables the right to be forgotten.**

**DP-ICL outperforms all previous DP-SGD methods on SST-2 benchmark (Table 5).** We also compare our results with current state-of-the-art differentially private stochastic gradient descent (DP-SGD) methods on **SST-2**. The results illustrate an improvement over earlier methods. For instance, the enhancement at $\varepsilon = 3$ is **1.2%**, which translates to an over **20%** reduction in relative error rate, thereby establishing a new SOTA in the field.[6] Moreover, by presenting the results with $\varepsilon = \infty$, we notice that our performance gains do not correlate to the advanced large language model, where our upper bound is lower than other methods. That means DP-ICL exhibits less sacrifice when achieving a differential privacy guarantee.

Table 5: **Comparison of DP-ICL and DP-SGD on the SST-2 Dataset.** Our DP-ICL method demonstrates significantly lower performance degradation under privacy constraints of $\varepsilon = \{3, 8\}$.

| Model | Method | $\varepsilon = 3$ (gap) | $\varepsilon = 8$ (gap) | $\varepsilon = \infty$ |
|---|---|---|---|---|
| RoBERTa-large (Liu et al., 2020) | DP-SGD (Li et al., 2022) | 93.04 (-3.16) | 93.81 (-2.39) | 96.20 |
| | DP-SGD (Yu et al., 2022) | – | 95.30[*](-1.10) | 96.40 |
| | DP-SGD (Bu et al., 2023) | 94.60 (-0.90) | 94.70 (-0.80) | 95.50 |
| | DP-SGD (He et al., 2023) | 94.23 (-1.97) | 94.87 (-1.33) | 96.20 |
| GPT-3 Davinci | DP-ICL (Ours) | **95.80**$_{0.21}$ (+0.07) | **95.83**$_{0.22}$ (+0.10) | 95.73 (4-shot) |

[*] Result present in (Yu et al., 2022) is $\varepsilon = 6.7$.

However, DP-ICL in Table 5 is capable of responding to a maximum of 10,000 queries, whereas DP-SGD has no such query limit and can answer an arbitrarily large number of queries. We defer to

---

[6]A minor discrepancy exists between our training data (sentence level) and the DP-SGD training data (phrase level) on the SST-2 dataset. Our training data is 10 times smaller than that of DP-SGD. However, the test data remains identical for both.

practitioners to evaluate these factors when selecting the most appropriate algorithm for real-world applications.

**Remark 2.** *We acknowledge that our DP-ICL is a private* prediction *framework, which does not allow for an infinite amount of queries. However, we stress that in our experiment, we consider up to 100,000 queries, which is arguably a huge number. Furthermore, in many real-world applications where ICL is helpful, in-context examples will often be updated regularly. For example, for healthcare data analysis, systems that analyze patient data for research or treatment optimization might update their learning models regularly with new data. Banks and financial institutions might use in-context learning for fraud detection, credit scoring, or personalized customer services. The dynamic nature of financial transactions could necessitate frequent updates to the learning models, aligning with a refreshed privacy budget.*

# D    DETAILS OF EXPERIMENTS SETUP

In this appendix, we present more details of our experiment setup, including the text classification task (Appendix D.1) as well as the language generation tasks (Appendix D.2).

## D.1    TEXT CLASSIFICATION

**Task.** Following Zhang et al. (2022), we employ the template summarized in Table 6 to carry out experiments on text classification tasks. We configure the logit bias to 100 via the GPT-3 API to ensure that the output token belongs to one of the predefined labels (e.g., positive and negative). We then select the label with the highest probability. We use a value of 0.0 for the temperature.

Table 6: The template prompts for our experiments.

| Dataset | Template | Labels |
|---------|----------|--------|
| SST-2 | Review: {text} Sentiment: {label} | Positive, Negative |
| Amazon | Title: {title} Review: {review} Sentiment {label} | Positive, Negative |
| AGNews | Article: {text} Answer: {label} | World, Sports, Business, Technology |
| TREC | Classify the questions based on whether their answer type is a Number, Location, Person, Description, Entity, or Abbreviation. Question: {text} Answer Type: {label} | Number, Location, Person, Description, Entity, Abbreviation |

**Dataset.** Our experiments are conducted on four datasets, the details of which are presented in Table 7. Given that the training data size for each dataset is fewer than 10,000, we set $\delta$ to $10^{-4}$.

Table 7: Information about the Text Classification Dataset

| Dataset | Task | # of classes | # of exemplars | # of test data | avg. length |
|---------|------|--------------|----------------|----------------|-------------|
| SST-2 | Sentiment cls. | 2 | 6,920 | 872 | 37.8 |
| Amazon | Sentiment cls. | 2 | 8,000 | 1,000 | 78.5 |
| AGNews | Topic cls. | 4 | 8,000 | 1,000 | 19.3 |
| TREC | Question cls. | 6 | 5,452 | 500 | 10.2 |

## D.2    LANGUAGE GENERATION

We then provide the detailed setup for document questions answering and dialog summarization.

### D.2.1    DOCUMENT QUESTIONS ANSWERING

**Task.** For document questions answering task, we use data from Privacy Preserving Federated Learning Document VQA (PFL-DocVQA) competition [7]. The primary objective is to create privacy-preserving approaches for fine-tuning multi-modal models, for both vision and language, to improve document understanding. These documents often contain sensitive or confidential information. As our research is centered on language generation, we skip the vision module and assume that accurate OCR tokens have been extracted from the document images. We leverage the language model to answer the posed questions, and an example of our evaluation is presented as follows:

> **Extracted OCR tokens from image:**
> Page,2,of,2,DUNORICE,REISSUE,Send,Payment,To:,WSIL-TV,Invoice,...
> **Question:** What is the identification number assigned to the invoice in the document?

---

[7]The web link is `https://benchmarks.elsa-ai.eu/?ch=2`

---

**Answer the question with short term:** 12970-2B

**Extracted OCR tokens from image:**
ORDER,WORKSHEET,Rep,Order#,10447675,Ver#,1,Status,New,Traffic,...
**Question:** Could you share the ID associated with the document?
**Answer the question with short term:**

---

**Dataset and Metrics.** We use the training data contains 221,329 data and evaluate the performance on 100 data that are randomly selected from validation dataset. Therefore, we set the $\delta$ for DP guarantee to $4 \times 10^{-6}$. We use three evaluation metrics in this task: ROUGE-1, BLEU, and Levenshtein similarity. Specifically, ROUGE-1 measures the overlap of unigrams between the generated and reference texts, and BLEU checks for matching words but considers them in n-grams. The Levenshtein similarity metric is derived from the Levenshtein distance, which measures the minimum number of single-character edits needed to transform one string into another.

**Model.** Due to the long OCR-extracted tokens, we avoided using the paid GPT API. We plan to conduct more extensive experiments if we have increased funding. For the current evaluation, we used the open-source model OpenLLaMA-13B Geng & Liu (2023).

### D.2.2 DIALOG SUMMARIZATION

**Task.** Then we provide the detailed experiment setup for dialog summarization, where we use the SAMsum dataset Gliwa et al. (2019). This dataset has conversations between people, some of which may be private. We use the prompt to guide the task, which is as follows:

---

**Dialogue:**
– Sandra: Are you sleeping?
– Matija: Nope
– Sandra: How so?
– Matija: It is Friday :D
– Sandra: So you are partying?
– Matija: File photo :D
– Sandra: You are in bed already?
– Matija: Of course :P
– Sandra: You are like old folks ;) What are your plans for tomorrow?
– Matija: Cleaning the apartment
– Sandra: Wow, nice
– Matija: Yea. Afterwards we are going to some restaurant, I have a lesson at 4 and we are meeting that bartender at 6
– ...

**Summarize the above dialogue:**
Matija is in his bed on Friday nights. Tomorrow he is cleaning the apartment and afterwards they are going to the restaurant to have a lesson at 4 and meet that bartender at 6.
**Dialogue:**
– Joyce: Check this out!
– Joyce: <link>
– Michael: That's cheap!
– Edson: No way! I'm booking my ticket now!!

**Summarize the above dialogue:**

---

When using keyword space aggregation by PTR, we change the *"Summarize the above dialogue:"* to *"Summarize the above dialogue with the following word suggestions:"*. When using keyword space aggregation by joint EM, we change the *"Summarize the above dialogue:"* to *"Summarize the above dialogue with the following word suggestions ranked by their frequency from high to low:"*.

**Dataset and Metrics.** We use the training data containing 14,732 data and evaluate the performance on 100 test data. We set the $\delta$ to $5 \times 10^{-5}$. Following He et al. (2023), we use ROUGE-1, ROUGE-2, and ROUGE-L as the evaluation metrics.

**Model.** We leverage the GPT-3 Davinci to conduct the experiments. Our methods can also apply to other large language models.

# E   ADDITIONAL EVALUATIONS ON TEXT CLASSIFICATION

In this appendix, we provide additional ablation studies for the text classification task. We discuss the impact of using DP-ICL with different queries (Appendix E.1) and models (Appendix E.2). We also examine how performance changes with varying numbers of in-context examples (Appendix E.4) and subsampling rates (Appendix E.5).

## E.1   EFFECTIVENESS OVER VARIOUS QUERIES

In this subsection, we conduct ablation experiments to learn the influence of varying the number of queries on performance, as illustrated in Figure 6. The outcomes for the TREC dataset were obtained using the GPT-3 Davinci model, whereas results for all other datasets employed the GPT-3 Babbage model. We keep $\varepsilon = 3$ constant and use 10 ensembles. Our observations reveal that the performance degradation resulting from an increase in the number of queries remains negligible, up to 10,000, with a decrease of less than 2%. Furthermore, it was observed that performance drops were more significant for the AGNews and TREC datasets compared to the remaining datasets. We hypothesize that this drops attributed to the property of multiple classes (i.e., 4 and 6) within the AGNews and TREC datasets.

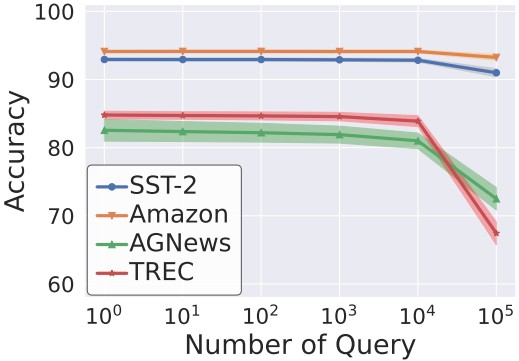

Figure 6: Performance across numbers of the query.

## E.2   EFFECTIVENESS OF VARIOUS MODELS

In this subsection, we explore various GPT-3 variants available through OpenAI's service. Specifically, we evaluate GPT-3 Ada (350M), GPT-3 Babbage (1.3B), GPT-3 Curie (6.7B), and GPT-3 Davinci (175B) on four text classification tasks, as detailed in Table 8. We configure the setting to be the same as our results in Table 2.

Table 8:  Results of DP-ICL for Text classification

| Dataset | Model | $\varepsilon = 0$ (0-shot) | $\varepsilon = 0$ (Pub.) | $\varepsilon = 1$ | $\varepsilon = 3$ | $\varepsilon = 8$ | $\varepsilon = \infty$ (Agg) | $\varepsilon = \infty$ |
|---|---|---|---|---|---|---|---|---|
| SST-2 | Ada | 86.24 | 80.16 | 73.31 | 75.10 | 75.38 | 72.29 | 70.92 |
| | Babbage | 86.58 | 89.45 | 91.97 | 92.83 | 92.90 | 92.87 | 91.89 |
| | Curie | 91.51 | 92.89 | 94.03 | 94.86 | 94.94 | 95.07 | 94.03 |
| | Davinci | 94.15 | 95.30 | 95.11 | 95.80 | 95.83 | 95.73 | 95.49 |
| Amazon | Ada | 92.00 | 84.90 | 89.68 | 90.26 | 90.32 | 90.70 | 88.51 |
| | Babbage | 93.80 | 91.60 | 93.83 | 94.10 | 94.12 | 94.10 | 93.58 |
| | Curie | 95.90 | 94.30 | 95.37 | 95.67 | 95.69 | 95.58 | 95.28 |
| AGNews | Ada | 37.50 | 40.00 | 65.60 | 70.51 | 71.22 | 71.72 | 60.23 |
| | Babbage | 52.60 | 61.80 | 75.49 | 81.00 | 81.86 | 82.22 | 68.77 |
| | Curie | 62.40 | 68.10 | 80.00 | 81.88 | 82.06 | 81.80 | 78.31 |
| TREC | Ada | 20.40 | 22.80 | 23.72 | 25.41 | 25.86 | 26.28 | 22.62 |
| | Babbage | 23.00 | 24.40 | 24.48 | 26.36 | 26.26 | 26.32 | 27.00 |
| | Curie | 29.20 | 29.80 | 37.95 | 41.96 | 42.50 | 42.24 | 36.57 |
| | Davinci | 79.60 | 80.60 | 73.31 | 83.86 | 84.53 | 85.92 | 79.08 |

We observe consistent performance improvements as the model becomes more advanced—indicated by increased parameters—across all datasets. For example, in the SST-2 dataset, zero-shot performance increased from 86.24% for Ada to 94.15% for Davinci. Correspondingly, our DP-ICL results also improved, rising from 73.31% to 95.11% with $\varepsilon = 1$ on the SST-2 dataset. This suggests that more advanced models in the future can further enhance DP-ICL performance. Interestingly, zero-shot performance can sometimes surpass in-context learning results, particularly for the Ada model on the Amazon dataset. This is caused by the influence of suboptimal in-context exemplars or limited in-context learning capabilities for small models. Therefore, estimating the utility of in-context learning (ICL) and zero-shot learning is essential before deploying DP-ICL.

We also present the results of four-shot predictions utilizing publicly available data in Table 8. Specifically, we apply data from the IMDB (Maas et al., 2011) dataset for the SST-2 and Amazon tasks, the AriseTV (chimaobi Samuel., 2022) dataset for the AGNews task, and the QQP (Wang et al., 2017) dataset for the TREC task, following the setting of (Duan et al., 2023b). Our analysis reveals that using public exemplars generally enhances performance compared to the zero-shot method, but this improvement is not uniformly observed across all datasets. The reason lies in the varied quality of public data, where low-quality public data can introduce inaccurate correlations between input and output mappings. On the other hand, DP-ICL with $\varepsilon = \{3, 8\}$ can consistently outperform the 4-shot prediction with public data on all settings except the SST-2 with the Ada model. Particularly, the improvement on the AGNews dataset is over 20%. It is also worth noting that in many domains, a high-quality, labeled public dataset may not be available, especially in specialized domains like healthcare, where in-context exemplars often require highly specific and sensitive data.

### E.3 EFFECTIVENESS OF MORE ADVANCED MODELS

In Table 9, we present the analysis of GPT-3.5[8] and GPT-4[9] across various datasets, using both 0-shot and 4-shot learning. The results indicate that the 4-shot approach yields consistently better performance than the 0-shot prediction, especially for AGNews and TREC. This trend underscores the importance of utilizing in-context learning to enhance model efficacy even for more advanced models in some scenarios.

Table 9: Zero-shot and Four-shot Results for more advanced models

| Dataset | GPT-3.5 (0-shot) | GPT-3.5 (4-shot) | GPT-4 (0-shot) | GPT-4 (4-shot) |
|---------|------------------|------------------|----------------|----------------|
| SST-2   | 95.06            | 95.18            | 93.23          | 94.15          |
| Amazon  | 96.20            | 96.60            | 95.90          | 96.00          |
| AGNews  | 76.20            | 84.40            | 83.80          | 86.10          |
| TREC    | 77.60            | 81.60            | 80.40          | 85.80          |

Table 10: Effectiveness of DP-ICL for more advanced models

| Dataset | GPT-3.5 (0-shot) | | GPT-4 (0-shot) | GPT-3.5 (4-shot) |
|---------|------------------|-------|----------------|------------------|
| AGNews  | 76.20            | 84.40 | 83.80          | 86.10            |
| TREC    | 77.60            | 81.60 | 80.40          | 85.80            |

### E.4 EFFECTIVENESS OVER THE NUMBER OF IN-CONTEXT EXEMPLARS.

In our primary evaluation in Table 2, we set our analysis to four in-context exemplars. In this ablation study, we extend our evaluation to include a broader range of in-context exemplars: 1, 2, 4, 8, and 16. For these experiments, we maintain an ensemble size of 10. Therefore, using more shots requires a higher subsampling rate, which increases noise in the voting histogram.

---

[8]Exact model name: `gpt-3.5-turbo-instruct`
[9]Exact model name: `gpt-4-1106-preview`

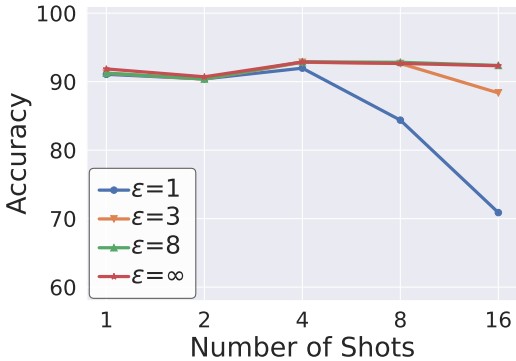

Figure 7: Performance across numbers of the in-context exemplars.

In Figure 7, our evaluation reveals that performance remains stable for $\varepsilon = \infty$. However, a significant performance degradation occurs for setups with 8 and 16 shots, experiencing drops of about 10% and 20%, respectively. This degradation is attributed to the introduction of additional noise.

## E.5 EFFECTIVENESS OVER SUBSAMPLING RATES

To better understand the cost-utility trade-off in our proposed DP-ICL, we investigate the impact of subsampling strategies in Fig. 8. To save the computation cost, our evaluation is limited to test sets comprising 100 data for two datasets: SST-2 and AGNews. Our empirical results indicate that a subsampling rate of around $0.5 \times 10^{-2}$ for the exemplars achieved satisfactory performance, which are 10 samples post-subsampling. We further evaluate the estimated cost associated with querying the GPT-3 Babbage API, as illustrated in Fig. 8 (Right). Notably, at a subsampling rate of $0.5 \times 10^{-2}$, the cost is only \$0.0945 for 100 predictions on the SST-2 dataset. Increasing the subsampling rate beyond this point does not significantly improve performance while resulting in higher costs.

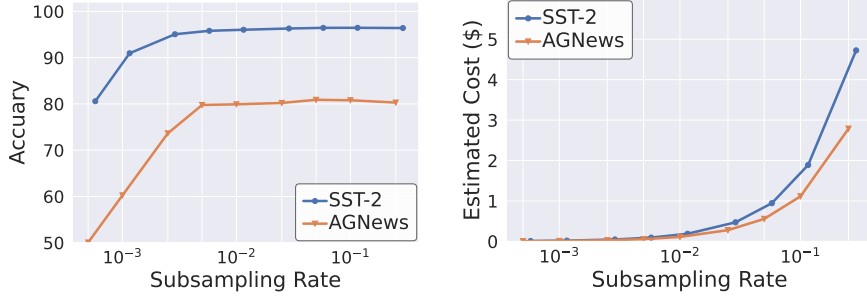

Figure 8: Left: Performance across the subsampling rate. Right: Estimated GPT-3 Babbage API Cost of predicting 100 test samples. The size of the training set for SST-2 is 6,920, and for AGNews, it is 8,000.

Furthermore, we also plot the cost and accuracy tradeoff in Figure 9 combining two figures in Figure 8. We found that using the GPT-3 Babbage API to predict 100 data points from the SST-2 and AGNews incurs a mere cost of approximately \$0.1 while achieving a comparable performance.

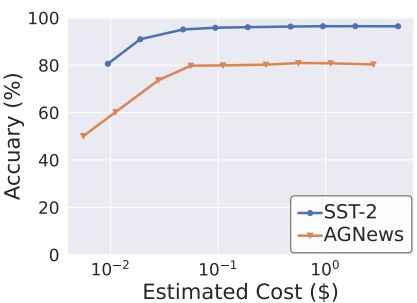

Figure 9: The trade-off between cost and accuracy for GPT-3 Babbage API.

## F    ADDITIONAL EVALUATIONS ON LANGUAGE GENERATION

In this appendix, we present ablation studies for language generation tasks. Results for the document question-answering task are in Appendix F.1, and those for dialog summarization are in Appendix F.2.

### F.1    DOCUMENT QUESTION ANSWERING TASK

**Performance across varying numbers of queries (Figure 10)** In document question answering tasks, we evaluate the keyword space aggregation via PTR method, as it consistently outperforms embedding space aggregation. Our analysis reveals that performance remains stable up to $10^6$ queries, allowing us to guarantee a strong performance for at least $10^5$ queries.

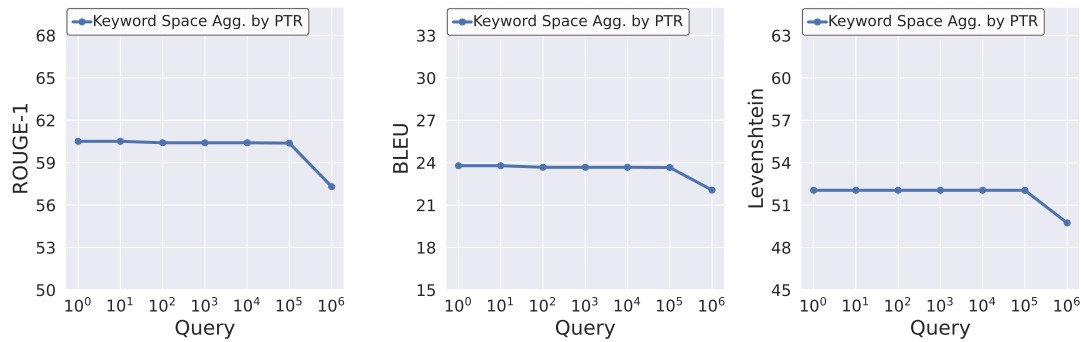

Figure 10: Ablation studies on varying numbers of queries.

### F.2    DIALOG SUMMARIZATION TASK

**Effectiveness across numbers of public candidates (Figure 11).** We study the performance of the embedding space aggregation method with varying numbers of public candidates. As anticipated, an increased number of public candidates leads to improved performance. The ideal scenario would involve maximizing the number of public candidates. However, a larger pool of candidates also needs more API queries, thereby increasing the overall cost.

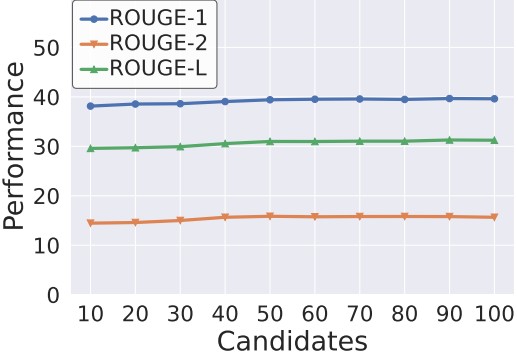

Figure 11: Performance across numbers of the public candidates (i.e., number of answers generated by zero-shot predictions) for embedding space aggregation method.

**Effectiveness across numbers of ensembles (Figure 12).** Here, we present the full results of ablation studies of the number of ensembles.

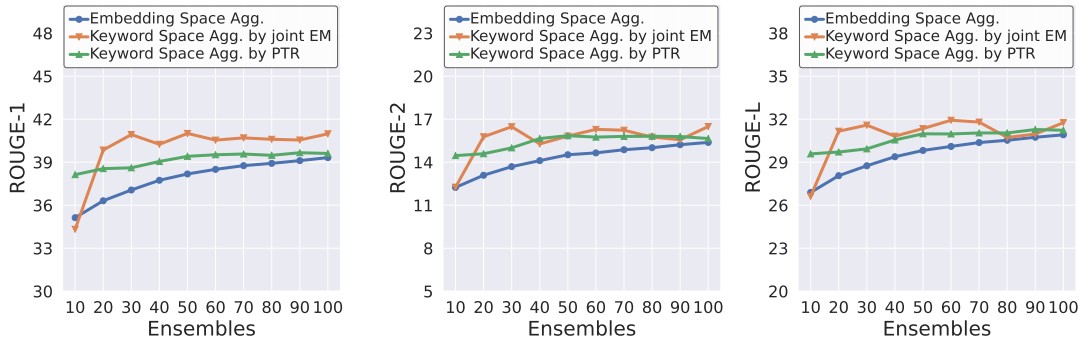

Figure 12: Ablation studies on varying numbers of ensembles. Full results of Figure 5(a)

**Effectiveness across numbers of queries (Figure 13).** Here, we present the full results of ablation studies of the number of queries.

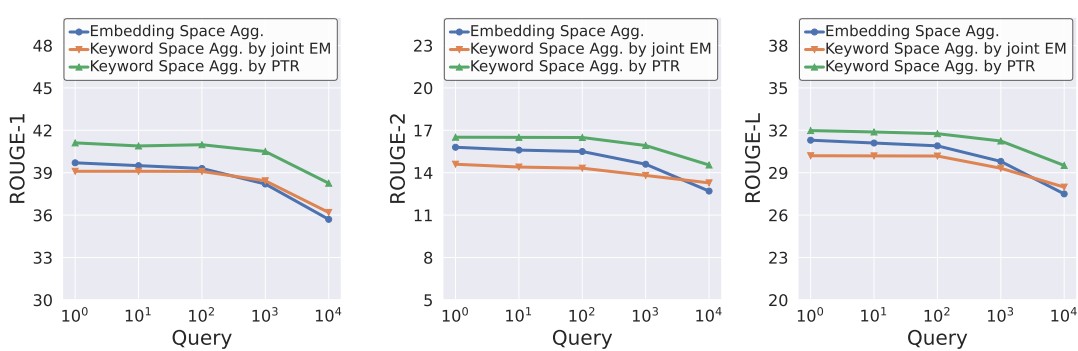

Figure 13: Ablation studies on varying numbers of queries. Full results of Figure 5(b)

### F.3 EFFECTIVENESS OF ADVANCED MODELS

In this subsection, we evaluated the zero-shot and four-shot prediction capabilities in more advanced models, specifically GPT-3.5 and GPT-4. Our findings reveal that utilizing four-shot predictions, which involve in-context learning with private exemplars, significantly enhances overall model performance. For instance, the improvement across all metrics on Document QA for GPT-4 is larger than 12%. This improvement trend is consistent with results from the dialog summarization task.

Table 11: Zero-shot and Four-shot Results for more advanced models

| Task | Prediction | GPT-3.5 (0-shot) | GPT-3.5 (4-shot) | GPT-4 (0-shot) | GPT-4 (4-shot) |
|---|---|---|---|---|---|
| Document QA | ROUGE-1 ↑ | 63.18 | 73.68 | 66.21 | 78.44 |
| | BLEU ↑ | 39.28 | 51.42 | 37.09 | 51.15 |
| | Levenshtein ↑ | 52.13 | 66.14 | 59.80 | 74.89 |
| Dialog Summ. | ROUGE-1 ↑ | 33.22 | 39.44 | 28.64 | 38.83 |
| | ROUGE-2 ↑ | 11.91 | 16.33 | 10.28 | 15.79 |
| | ROUGE-L ↑ | 25.08 | 31.21 | 21.95 | 30.41 |

### F.4 TRADE-OFF BETWEEN COST AND ACCURACY

In this subsection, we present the trade-off between cost and accuracy for dialog summarization task. Here, we used the price [10] of the updated version of GPT-3 Davinci to estimate the final cost of summarizing 100 dialogs. We present the results in Figure 14, where we vary the ensemble number

---

[10]GPT-3 Davinci is going to be deprecated on Jan 4th 2024 and the suggested replacement by OpenAI is `gpt-3.5-turbo-instruct`

from 10 to 100. Specifically, the cost of the ESA is calculated by both multiple ensemble queries and the generation of potential candidates (0-shot predictions). For KSA, the cost also involves multiple ensemble queries plus an additional 0-shot prediction for constructing the final sentence from keywords. We set the candidate number to 100 for ESA. We observe that ESA by joint EM can achieve comparable performance with only ∼\$2.

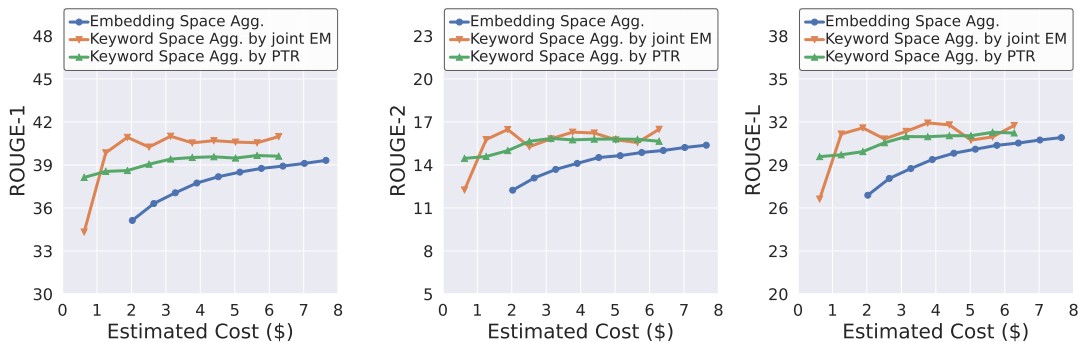

Figure 14: The trade-off between cost and accuracy for dialog summarization task.

## G    EXAMPLES OF ESA OUTPUT IN DIALOG SUMMARIZATION TASK

In this section, we offer some qualitative examples of zero-shot candidates and the final outputs selected by embedding similarity. It is observed that the final outputs demonstrate not only fluency but also a heightened similarity to the ground truth compared to the zero-shot candidates.

---

**Dialogue:**
– Sybille: Hello, i'm trying to make my online registration for the flight AF3581 from Minneapolis to Paris the 29th, but i only get an error message. Here is my file number NRTTU and my flying blue number XXXX. Thanks for you help.
– Air France: Hello Sybille, we acknowledge receipt of your message and return to you as soon as possible.
– Air France: After checking your flight is operated by Delta Airlines for departure from the United States. Online registration works from the Delta site.
– Sybille: Hello again , i still have trouble for my son' s online registration. I'm really angry as it cost me more than 40 euros for 20 minutes on your short number 3654, and still i don't have any answer. He couldn't get his flight yesterday, so i asked for it to be postpone. I twice gave my credit number but still the ticket is not delivered.
– Air France: Of course, i give the file to an agent.
– Air France: Please, be awared it could take 24 hours
– Sybille: I can't wait so long. My son is only 16 year old and he has to sleep in the airport, with no mean to leave his luggage as he has no ticket.
– Air France: Hello Sybille, we're very sorry . We could offer you a departure on flight DL140 from Minneapolis to Paris. We'll return to you.
– Sybille: My phone number is XXXX, please call me back as soon as possible
– Sybille: Hello, i was again on line with people from the short number. They told me that they've done everything possible but my payment can't be accepted because the flight is operated by Delta. Please consider a young stuck in US with no possibilites for coming back just because of Delta and Air France sharing code, can't share payment.
– Air France: We'll get in touch with Delta US and return to you quickly
– Sybille: thanks a lot
– Air France: We just send you by email the new ticket for you son. We remain at your disposal.
– Sybille: Thanks a lot, you're more efficient than the hotline.

---

**Zero-shot summarization candidates:**
• Sybille contacted Air France trying to make her son's online registration. After recognizing the flight was operated by Delta, Air France suggested the customer to complete the Al online registration on the Delta website. Sybille then proceeded to point out her dissatisfaction regarding the long time she spent on the hotline without getting any solution. Air France then offered her a new flight DL140 from Minneapolis to Paris, and subsequently sent her a new ticket for her son by email. Sybille thanked Air France for their help.
• Sybille was trying to make an online registration for her son's flight from Minneapolis to Paris but kept getting an error message. She contacted Air France and was told to register online with Delta Airlines. After explaining her son was stuck in the US with no means to leave, Air France contacted Delta US and quickly sent her an email with the new ticket for her son. Sybille thanked Air France for their efficiency.
• Sybille was attempting to make an online registration for a flight with Air France, but encountered an error message. Air France told her the flight was operated by Delta Airlines, who she should use to do the registration. The customer became frustrated as she was charged for a phone call, yet still had not received a successful registration. Air France offered a new ticket for her son via email and the customer thanked them for their efficiency.
• Sybille is trying to make her son's online registration for the flight AF3581 from Minneapolis to Paris on the 29th, but she only gets an error message. She contacts Air France customer service, who informs her that the flight is operated by Delta and so she must use the Delta online registration system. When Sybille is still having trouble, she expresses her frustration at the costly and time consuming customer service hotline. Air France then contacts Delta and sends a new ticket for her son by email.

- Sybille is trying to make an online registration for a flight from Minneapolis to Paris for her son, but she is encountering issues. Air France offers an alternative flight and eventually sends a new ticket via email. Sybille thanks Air France for their help.
- A customer was unable to make an online registration for their son's flight from Minneapolis to Paris and forked out 40 euros for a 20 minute phone call. After Air France checked the flight flight, the customer was informed that the registration must be done through the Delta website. Air France offered the customer an alternative flight option and eventually sent the customer an email with a new ticket for their son. The customer thanked Air France for their efficiency.
- Sybille is trying to make an online registration for her son's flight from Minneapolis to Paris on the 29th, but is running into trouble. Air France acknowledges her concerns and directs her to the Delta Airlines website in order for her to make the registration. After Sybille expresses her anger over the delay and her son sleeping at the airport without any mean to leave his luggage, Air France offers her a departure on flight DL140 from Minneapolis to Paris. She provides Air France with her phone number for them to call her back, and after getting in touch with Delta US, Air France emails Sybille a new ticket for her son.
- Sybille contacted Air France to make an online registration for her son for a flight from Minneapolis to Paris, but encountered errors. Air France suggested that the registration should be done from the Delta site, but Sybille was unable to register due to payment issues.Air France sent an agent to help and offered an alternate flight. Finally, they sent an email with the new ticket for Sybille's son and thanked them for their help.
- Sybille needed help with registration for a flight operated by Delta Airlines and encountered problems with online registration, payment, and lack of response via the hotline. Air France responded by looking into the information and offering a different flight option, which they later emailed the ticket for. In the end, Sybille thanked them for their efficiency.
- ...

**Selected candidate:**
Sybille is trying to make an online registration for a flight from Minneapolis to Paris for her son, but she is encountering issues. Air France offers an alternative flight and eventually sends a new ticket via email. Sybille thanks Air France for their help.
**Ground Truth:**
Sybille is angry, because she cannot make a reservation for the flight from Minneapolis to Paris. Air France managed to solve the issue.

