# OpenReview forum: "Privacy-Preserving In-Context Learning for Large Language Models"
_ICLR.cc/2024/Conference — ICLR 2024 poster_

### Official Review · Reviewer_ZNqV · 2023-10-15

**Soundness:** 3 good
**Presentation:** 4 excellent
**Contribution:** 4 excellent
**Rating:** 8
**Confidence:** 4

**Summary:**

The paper proposes a new paradigm for privatizing in-context learning tasks called “DP-ICL”, which achieves a strong utility-privacy trade-off and a comparable performance as the non-private counterpart. The authors demonstrate the effectiveness of DP-ICL on text classification and language generation tasks, and show that it offers a promising overall paradigm for applying ICL in a privacy-preserving way.

**Strengths:**

- This paper is well-written and well-organized.
- The proposed paradigm is novel and is a novel paradigm for privatizing ICL tasks.
- The authors provide a clear and concise explanation of the DP-ICL paradigm. The figures and captions are easy to understand.
- Experiments on 4 benchmark datasets are extensive and sufficiently demonstrated the promise of the approach towards trustworthy usage of large language models.

**Weaknesses:**

- Limitations can be discussed in more details. For example, “a better-trained embedding-to-text model may yield further improvements in the model performances.” Then what can be the potential candidate models here and why did the authors not adopt these models?

**Questions:**

- How do you see the DP-ICL paradigm being applied in real-world scenarios, and what are some of the challenges that would need to be addressed?
- What are some of the limitations of the DP-ICL paradigm, and how might they be addressed in future research?

---

> ### Author Response · Authors · 2023-11-17
>
> We thank the reviewer for the positive feedback and for acknowledging our paper’s novelty. We have provided detailed answers as follows:
>
> > **Q1. [Future work for ESA]**  *“For example, “a better-trained embedding-to-text model may yield further improvements in the model performances.” Then what can be the potential candidate models here and why did the authors not adopt these models?”*
>
> **A:** For the ESA method, the overall performance highly depends on the embedding-to-text mapping. At the time of submission, a dedicated model/method for embedding reconstruction wasn't available, leading us to approximate the embedding-to-text mapping using zero-shot candidates. A very recent work that appears after our paper submission [1] introduces advanced techniques that could directly reconstruct text from embedding. We plan to explore these techniques in future research to improve the accuracy of our ESA method.
>
> [1] Morris et al. Text Embeddings Reveal (Almost) As Much As Text. ArXiv 2023.
>
> > **Q2. [Real-world Applications]**  *How do you see the DP-ICL paradigm being applied in real-world scenarios, and what are some of the challenges that would need to be addressed?*
>
> **A:** The DP-ICL paradigm has significant potential for real-world application, particularly for third-party companies managing confidential data. For instance, consider a healthcare institution with a repository of private clinical records. This organization could implement DP-ICL by hosting LLMs through an API endpoint. It would enable external users to query the LLM and receive responses based on the sensitive data stored while safeguarding privacy.
> However, this application raises several challenges. Firstly, maintaining the balance between data privacy, the utility, and the computational cost of the LLM's responses presents a complex challenge. Practitioners should examine the tradeoffs before deploying DP-ICL.
> Furthermore, many real-world applications first retrieve useful information from a large database and then apply ICL [2, 3], also known as retrieval-augmented-generation (RAG). Our current design leverages Possion subsampling to select the exemplars. Extending our DP-ICL framework to other retrieval techniques could be an interesting future work.
>
>
> > **Q3. [Limitations & Future Work of DP-ICL]**  *“What are some of the limitations of the DP-ICL paradigm, and how might they be addressed in future research?”*
>
> **A:**
> Besides the limitation and future directions we discussed in Q1 and Q2, another limitation lies in that DP-ICL cannot handle an infinite amount of queries. In our experiments, we have shown that our proposed algorithms can handle up to 100,000 queries within the prespecified privacy budget, which is already quite high. However, it is still possible to further improve it from multiple perspectives. For example, a tighter privacy accounting method for PTR and Exponential mechanism could allow more queries under the same privacy budget. Another idea is to construct a dynamic database for ICL such that private data is regularly updated and the privacy budget can be refreshed.
>
> We look forward to engaging in further discussions with you.
>
> [1] Morris et al. Text Embeddings Reveal (Almost) As Much As Text. arXiv 2023.
>
> [2] Veen et al. Clinical text summarization: Adapting large language models can outperform human experts. ArXiv 2023.
>
> [3] Liu et al. LlamaIndex.(https://github.com/run-llama/llama_index)

---

> ### Author Response · Authors · 2023-11-21
> **Looking forward to hearing from you**
>
> Dear Reviewer ZNqV,
>
> Thank you again for the positive feedback and thoughtful comments. We would love to ask if you have any further comments or suggestions for our paper.
>
> Best,
>
> Authors

---

> ### Comment · Reviewer_ZNqV · 2023-11-21
> **Thanks for the response**
>
> Thanks for your responses, which have resolved all of my concerns. I tend towards the acceptance of this work

---

> > ### Author Response · Authors · 2023-11-21
> > **Thank you**
> >
> > Thank you once again for supporting our paper.

---

### Official Review · Reviewer_2dnr · 2023-10-31

**Soundness:** 3 good
**Presentation:** 2 fair
**Contribution:** 2 fair
**Rating:** 6
**Confidence:** 2

**Summary:**

The author proposed a new Differentially Private In-Context Learning framework (DPICL) to minimize privacy risk in in-context learning. The main idea is to leverage the private “sample-and-aggregate” paradigm. For text-classification, the paper proposed to report the noisy max with gaussian noise; For text generation, the paper proposed to use embedding space aggregation or keyword space aggregation.

**Strengths:**

The proposed methods are intuitive and in the experimental section, authors find DP-ICL achieves a comparable performance with non-private ICL.

Thorough experimental studies ranging from text classification, question answering, and summarization. In particular, the proposed framework can be applied to QA and summarization tasks which are complex and challenging tasks.

**Weaknesses:**

It is a little confusing on how the author analyzed the sensitivity and how are the neighboring database is defined.

How much extra cost (monetary and privacy budget) does the proposed framework incurs? It would be great if the author can include some insights on the cost vs accuracy trade off.

Is comparing 4-shot vs 0-shot a fair comparison? In the related work section, the author mentioned work using examples from public dataset. Perhaps a better comparison is between private 4-shot using examples from the training data vs. 4-shot using examples from public data.

**Questions:**

See weakness.

---

> ### Author Response · Authors · 2023-11-17
>
> We thank the reviewer for recognizing the contributions of our paper and offering constructive suggestions. We have addressed your comments in the subsequent paragraph.
>
> > **Q1. [how the sensitivity is analyzed, and how the neighboring database is defined.]**
>
> **A:**  For **how the sensitivity is analyzed**: The comprehensive privacy analysis for each DP algorithm proposed in our paper is detailed in Appendix B. Specifically, the sensitivity of text classification algorithm is analyzed in the proof of Theorem 3. The sensitivity of ESA algorithm is analyzed in the proof of Theorem 4. For KSA via Joint EM, the sensitivity is analyzed in the proof of Theorem 9. For KSA via PTR, the sensitivity of Algorithm 6 (FindBestK) is provided in Theorem 10. PTR paradigm adds randomness in a data-dependent way, and the privacy analysis is provided in Theorem 11. **We have added an outline at the beginning of Appendix B to help the readers find the privacy analysis for each algorithm.**
>
> For **how the neighboring database is defined**: Throughout the entire paper, we use add/remove as the neighboring relation. We have modified our paper and stressed this point in the Background section.
>
> Additionally, for calculating the final privacy parameter $\varepsilon$, we use RDP- and PRV-based privacy accountant, which computes the privacy loss numerically, and the final privacy loss has no closed-form expressions. Specifically:
> - **For the experiments on text classification and embedding space aggregation (ESA)**, we use Privacy Loss Random Variable (PRV) accountant [1] because we are composing (subsampled) Gaussian mechanisms. The modern PRV-based privacy accountant computes the privacy loss numerically, and hence, the final privacy loss has no closed-form expressions. We stress that PRV accountant technique is a standard practice in differential privacy literature nowadays due to the tight privacy bound it can compute. We detailed the use of the PRV accountant in the last paragraph of Appendix B.2 of our submission. To further highlight this paragraph, **we have converted that paragraph into Remark 1, and we have added a small paragraph in the main text to stress the use of PRV accountant and point the reader to Remark 1**.
> - **For the experiments on keyword space aggregation (ESA)**, we are composing (Subsampled) Propose-Test-Release (PTR) paradigm and Exponential Mechanism (EM). Since the PRV is unknown for either Propose-Test-Release or EM, PRV accountant is not applicable here and we instead use the tools of *Renyi DP* and *approximate Renyi DP* for privacy accounting. Similar to PRV-based accountant, RDP-based accountant also compute the privacy loss numerically, and hence the final privacy loss has no closed-form expressions. The detailed (approximate) RDP analysis for the PTR paradigm and Joint EM can be found in Theorem 9 to 12. **We have added a small paragraph in the main text to stress the use of RDP accountant and point the reader to Appendix B.3**.
>
>
> > **Q2. [Money Cost vs Accuracy Trade-off]** *“How much extra cost (monetary and privacy budget) does the proposed framework incur? It would be great if the author can include some insights on the cost vs accuracy trade-off.”*
>
> **A:** We appreciate the reviewer's valuable suggestion. **We have incorporated additional figures about cost-performance tradeoffs into the paper.**
>
> For text classification tasks, the cost is determined by the number of output ensembles. If we use 10 ensembles of 4-shot predictions, it leads to a roughly 10-fold cost increase over plain 4-shot prediction. The result is shown in this [figure](https://ibb.co/BfmbdZT), where we vary the number of ensembles to adjust the trade-off between the cost and accuracy. We found that using the GPT-3 Babbage API to predict 100 data points from the SST-2 and AGNews incurs a mere cost of approximately $0.1 while achieving a comparable performance. The implementation details can be found in **Appendix E.5**.
>
> For language generation tasks, the cost differs due to specific algorithm design. For ESA, the cost includes both ensemble queries and the generation of potential candidates (0-shot predictions). For KSA, the cost involves multiple ensemble queries plus an additional 0-shot prediction for reconstructing the sentence from keywords. The cost-accuracy trade-off for the dialog summarization task is shown in this [figure](https://ibb.co/KmxbW5N ). Here, we vary the number of ensembles from 10 to 100. As we can see, for KSA by Joint EM, conducting 100 times of dialog summarizations only requires less than $2 for a comparable utility. The implementation details are presented in **Appendix F.4**.

---

> ### Author Response · Authors · 2023-11-17
>
> >**Q3. [4-shot Prediction with public data]** *“Perhaps a better comparison is between private 4-shot using examples from the training data vs. 4-shot using examples from public data.”*
>
> **A:** Thanks for the valuable suggestion. Accordingly, we follow the setup in [1] by using different datasets as publicly available data for classification tasks. The result is shown in the following table (the implementation details are in **Appendix E.2**). As we can see from the figure, using public exemplars generally enhances performance compared to the zero-shot method; however, this improvement is not uniformly observed across all datasets. The variation may be attributed to the varying quality of the public dataset, where low-quality public data can introduce inaccurate correlations between input and output mappings. On the other hand, DP-ICL with $\varepsilon = \{3,8\}$ can consistently outperform the 4-shot prediction with public data on all settings except the SST-2 with the Ada model. Particularly, the improvement on the AGNews dataset is over 20%.
>
> It is also worth noting that in many domains, a high-quality, labeled public dataset may not be available, especially in specialized domains like healthcare, where **in-context exemplars often require highly specific and sensitive data**.
>
> [1] Duan et al. Flocks of Stochastic Parrots: Differentially Private Prompt Learning for Large Language Models. NeurIPS 2023.
>
>
> | Dataset | Model  | $\varepsilon$ = 0 (0-shot) | $\varepsilon$  = 0 (4-shot Pub.) | $\varepsilon$  = 3 (DP-ICL) | $\varepsilon$  = 8 (DP-ICL)|
> |---------|--------|----------------|--------------|-------|-------|
> | SST-2   | Ada    | 86.24          | 80.16        | 75.10 | 75.38 |
> |         | Babbage| 86.58          | 89.45        | 92.83 | 92.90 |
> |         | Curie  | 91.51          | 92.89        | 94.86 | 94.94 |
> |         | Davinci| 94.15          | 93.30        | 95.80 | 95.83 |
> | Amazon  | Ada    | 92.00          | 84.90        | 90.26 | 90.32 |
> |         | Babbage| 93.80          | 91.60        | 94.10 | 94.12 |
> |         | Curie  | 95.90          | 94.30        | 95.67 | 95.69 |
> | AGNews  | Ada    | 37.50          | 40.00        | 70.51 | 71.22 |
> |         | Babbage| 52.60          | 61.80        | 81.00 | 81.86 |
> |         | Curie  | 62.40          | 68.10        | 81.88 | 82.06 |
> | TREC    | Ada    | 20.40          | 22.80        | 25.41 | 25.86 |
> |         | Babbage| 23.00          | 24.40        | 26.36 | 26.26 |
> |         | Curie  | 29.20          | 29.80        | 41.96 | 42.50 |
> |         | Davinci| 79.60          | 80.60        | 83.86 | 84.53 |
>
>
> We hope the additional experiments and justifications can address all your concerns. We look forward to engaging in further discussions.

---

> ### Author Response · Authors · 2023-11-21
> **Looking forward to hearing from you**
>
> Dear Reviewer 2dnr,
>
> We are deeply grateful for your insightful and constructive comments. We hope we have properly addressed your concerns in our revisions and would love to ask if you have any additional comments or questions.
>
> Best,
>
> Authors

---

> > ### Comment · Reviewer_2dnr · 2023-11-21
> >
> > Thank you for the detailed response.
> > I have slightly raised my score.

---

> > > ### Author Response · Authors · 2023-11-21
> > > **Thank you**
> > >
> > > We deeply appreciate your feedback and thanks again for your help.

---

### Official Review · Reviewer_3Mai · 2023-11-01

**Soundness:** 3 good
**Presentation:** 3 good
**Contribution:** 3 good
**Rating:** 6
**Confidence:** 4

**Summary:**

This paper focuses on the problem of private data leakage of in context examples, during few-shot prompting (in-context learning). A new operation mode of LLMs is to ask a model to perform the task, and provide some examples for it, instead of actually performing any fine-tuning or changing the model parameters. In such use-cases, if the in-context examples have private information in them, there is a chance that it leaks to the output. To prevent this leakage, the authors propose differentially private in context learning methods for classification and generation.

These methods are all based on some form of noisy voting/decoding. The classification is directly using noisy max, but for the generation task the authors propose 3 different methods, to solve the problem of high dimensionality. They propose three different methods, one where the generation is mapped to embedding space, noisier and then mapped back to sentence space. The other two methods do noisy generation of keywords, and then uses a language model to generate text based on that.

The authors evaluate their proposed method on numerous tasks/datasets and compared to non-private baselines.

**Strengths:**

1. The problem the authors are studying is timely, as practitioners are not really fine-tuning models any more and rely on prompting more and more, therefore solutions like DP-SGD are not relevant.


2. The paper is very well-written and flows really well. The visuals are well-crafted and help the understanding of the paper a lot.

3. I like how thorough the paper is, in terms of the cases they study: classification and generation, and I also like the workarounds proposed for generation to limit the space. They are novel and also address the problem in a suitable way. Pairing up the keyword aggregation method with PTR was very interesting to me.

**Weaknesses:**

1. Limited number of queries: as there is budget expenditure per query, the model deployed with DP-ICL can only be queried a limited number of times, making the method/data unusable after the budget is finished. This is not a huge concern in scenarios where there is temporal data available, and the in-context examples would get updated regularly. However, it could be a problem during deployment for long term.

2. Zero-shot performance is already really high, would probably be even higher for GPT3.5 or GPT4, which are not reported in the paper.

3. There are no existing measures or proof of concept attacks that show private in-context data leakage.

**Questions:**

1. What are some real-world applications in which the authors think DP-ICL could be applicable, i.e. where the ICL few-shot samples are private, and using them is absolutely necessary, as for most of the existing tasks in the paper zero-shot performance is already high.


2. Do the authors have any measure of fluency for the generated text, specially for the embedding aggregate method? Human evaluations are usually performed in NLP generation tasks, human evaluations are usually performed to see how fluent the generated text is, and I wonder how the transformation of sentence to and from embedding impacts its fluency.



Minor comment:

The Min et al. 2022 reference is duplicated in the references (there are two versions of it)

---

> ### Author Response · Authors · 2023-11-17
>
> We thank the reviewer for the positive comments and for providing thoughtful feedback.
>
> > **Q1. [Private Prediction vs Private Learning]** *“as there is budget expenditure per query, the model deployed with DP-ICL can only be queried a limited number of times, making the method/data unusable after the budget is finished. This is not a huge concern in scenarios where there is temporal data available, and the in-context examples would get updated regularly. However, it could be a problem during deployment for long term.”*
>
> **A:**
> We acknowledge that our DP-ICL is a private *prediction* framework, which does not allow for an infinite amount of queries. However, we stress that **in our experiment, we consider up to 100,000 queries, which is arguably a huge number of queries for a single entity to perform.** There are two main considerations behind our choice of private *prediction* framework:
>
> **Compared with private learning, private prediction is more suitable for privatizing in-context learning.** In **Appendix C**, we discussed the pros and cons of private prediction compared with private learning. DP-SGD and PATE are the two mainstream techniques used in the literature of private learning. However, DP-SGD is clearly not able to extend to the setting of in-context learning. We also considered PATE, i.e., privately label a publicly available dataset and then use the newly labeled data pairs as demonstrations. Although viable for private learning, **the framework of PATE relies on the availability of a high-quality, unlabeled public dataset**. This assumption is often problematic in specialized domains like healthcare, where **in-context demonstrations often require highly specific and sensitive data**.
>
> **In many real-world applications where ICL is helpful, in-context examples will often be updated regularly.** For example, for healthcare data analysis, systems that analyze patient data for research or treatment optimization might update their learning models regularly with new data. Banks and financial institutions might use in-context learning for fraud detection, credit scoring, or personalized customer services. The dynamic nature of financial transactions could necessitate frequent updates to the learning models, aligning with a refreshed privacy budget.
>
> Based on these considerations, we decided to privatize ICL as a private prediction rather than private learning. We thank the reviewer for the valuable question, and we have incorporated the above discussion into Appendix C.

---

> ### Author Response · Authors · 2023-11-17
>
> > **Q2. [Zero-shot Performance of GPT-3.5 and GPT-4 & Necessity of DP-ICL]** *“Zero-shot performance is already really high, would probably be even higher for GPT3.5 or GPT4, which are not reported in the paper.”* *“What are some real-world applications in which the authors think DP-ICL could be applicable, i.e. where the ICL few-shot samples are private, and using them is absolutely necessary, as for most of the existing tasks in the paper zero-shot performance is already high.”*
>
> **A:** We have conducted the experiments for the zero-shot settings of GPT-3.5 and GPT-4 in the following tables (these additional results have been incorporated into **Appendix E.3 and F.3**).
>
> As we can see, while these models’ zero-shot performance is indeed high on simpler datasets such as SST-2 and Amazon, they still have significant space for improvement on more complex datasets such as AGNews, TREC, and the two language generation tasks. On the other hand, we observe that few-shot learning improves performance across **all** tasks. For instance, the improvement across all metrics on Document QA for GPT-4 is larger than 12%. **This experiment result demonstrates the necessity of DP-ICL,** as the effectiveness of DP-ICL correlates directly with the utility of few-shot learning, which is generally better than zero-shot learning.
>
> **Real-world applications for DP-ICL:** DP-ICL is most relevant to scenarios where the task requires domain-specific knowledge while privacy is important, as exemplified by the document Question Answering (QA) task from a recent competition focusing on privacy. On this task, zero-shot learning’s performance is relatively low compared to few-shot learning due to its domain-specific nature. Moreover, this competition's dataset notably includes sensitive information like company names (for more information, please refer to this [link](https://benchmarks.elsa-ai.eu/?ch=2)). We posit that DP-ICL holds significant potential for deployment in companies like financial and healthcare institutions, where they possess sensitive data. These institutions could utilize DP-ICL to host their own APIs, thereby assisting their users. A concrete example of this application is the recent introduction of [GPTs](https://openai.com/blog/introducing-gpts), where directly hosting an API with proprietary prompts has become feasible.
>
>
>
>
>
> | Dataset | GPT-3.5 (0-shot) | GPT-3.5 (4-shot) | GPT-4 (0-shot) | GPT-4 (4-shot) |
> |---------|------------------|------------------|----------------|----------------|
> | SST-2   | 95.06            | 95.18            | 93.23          | 94.15          |
> | Amazon  | 96.20            | 96.60            | 95.90          | 96.00          |
> | AGNews  | 76.20            | 84.40            | 83.80          | 86.10          |
> | TREC    | 77.60            | 81.60            | 80.40          | 85.80          |
>
> | Task          | Prediction | GPT-3.5 (0-shot) | GPT-3.5 (4-shot) | GPT-4 (0-shot) | GPT-4 (4-shot) |
> |---------------|------------|------------------|------------------|----------------|----------------|
> | Document QA   | ROUGE-1 ↑  | 63.18            | 73.68            | 66.21          | 78.44          |
> |               | BLEU ↑     | 39.28            | 51.42            | 37.09          | 51.15          |
> |               | Levenshtein ↑ | 52.13        | 66.14            | 59.80          | 74.89          |
> | Dialog Summ.  | ROUGE-1 ↑  | 33.22            | 39.44            | 28.64          | 38.83          |
> |               | ROUGE-2 ↑  | 11.91            | 16.33            | 10.28          | 15.79          |
> |               | ROUGE-L ↑  | 25.08            | 31.21            | 21.95          | 30.41          |

---

> ### Author Response · Authors · 2023-11-17
>
> > **Q3. [Limited measures of in-context data leakage]** *“There are no existing measures or proof of concept attacks that show private in-context data leakage.”*
>
> **A:** We thank the reviewer for the valuable comment, and we stress that the privacy risks of in-context learning (and, more generally, the LLM’s prompt) have been studied and demonstrated in many prior works. For example, [1] studied the privacy leakage of private information via ICL in the presence of privacy-preserving prompts. Furthermore, [2] developed a membership inference attack targeted at ICL, which can potentially expose whether a particular record was part of the training data. In addition, [3] has shown that Bing's secret prompt can also be extracted. On the other hand, our work is the **first** to explore the provable defense of these privacy attacks on ICL.
>
> Furthermore, we stress that anticipating and mitigating risks before they occur is crucial. Our work contributes to this effort by proposing DP-ICL, aiming to safeguard data privacy to ICL before any potential attacks occur.
>
>
>
> > **Q4. [Fluency measurement of DP-ICL]** *“Do the authors have any measure of fluency for the generated text, specially for the embedding aggregate method?.”*
>
> **A:** We thank the reviewer for suggesting measuring the fluency of generated text for the ESA method. We have included some qualitative examples of ESA sentences in **Appendix G**, demonstrating the fluency of the final output. Since we reconstructed the embedding by **selecting from the generated zero-shot candidates, the final output sentence is as fluent as zero-shot/few-shot predictions**.
>
>
> **Comment: [The Min et al. 2022 reference is duplicated]**
> Thanks for pointing out the typo, and we have revised it in the updated version.
>
> We thank you again for the thoughtful comments and suggestions. Please let us know if you have any further questions.
>
> [1] Wang et al. Decodingtrust: A comprehensive assessment of trustworthiness in gpt models. NeurIPS 2023.
>
> [2] Duan et al. On the Privacy Risk of In-context Learning. TrustNLP 2023.
>
> [3] Liu. https://twitter.com/kliu128/status/1623472922374574080.
>
> [4] Morris et al. Text Embeddings Reveal (Almost) As Much As Text. arXiv 2023.

---

> ### Author Response · Authors · 2023-11-21
> **Looking forward to hearing from you**
>
> Dear Reviewer 3Mai,
>
> Thank you for your insightful and constructive feedback. We have attempted to address each of your concerns in a thorough manner. If you have any further questions or comments, we are eager to discuss them with you.
>
> Best,
>
> Authors

---

> > ### Comment · Reviewer_3Mai · 2023-11-22
> > **rebuttal ack**
> >
> > I thank the authors for their response.
> >
> > I appreciate the new results, and thank the authors for it! I just have one question, is the first table showing the 4shot w/ DP-ICL? or normal ICL? If it is DP-ICL, it is convincing and I will increase the score. also, what is the epsilon value if it is?

---

> ### Author Response · Authors · 2023-11-22
>
> > **Q. Is the first table showing the 4-shot w/ DP-ICL? or normal ICL**
>
> **A:** We appreciate your further question and apologize for the confusion. The first table in our previous response shows the performance of normal zero-shot and four-shot predictions. The main message of these experiments is to demonstrate that **ICL is necessary for many tasks, especially for challenging tasks**. Since the effectiveness of DP-ICL is dependent on ICL, we expect that DP-ICL is also meaningful and necessary.
>
> Based on your suggestion in the last reply, we have additionally conducted DP-ICL experiments overnight to support our claim. Due to the limited time, we experimented with GPT3.5 on AGNews and TREC to demonstrate the effectiveness of DP-ICL. The table shown below includes the results of DP-ICL with $\varepsilon \in \{3, 8\}$. Additionally, we also include “4-shot agg.” which represents the aggregate predictions from ten 4-shot prediction ensembles. Notably, DP-ICL achieves an approximate 10% improvement over zero-shot predictions in AGNews. Similar improvements can also be observed in the TREC dataset. Additionally, we find that adding DP noise might not degrade the performance shown in AGNews. We conjecture that this is due to the regularization effect of additional random noise. However, the significant performance gap between DP-ICL and zero-shot performance can already illustrate the meaningfulness of DP-ICL. Therefore, our results support that **DP-ICL is highly meaningful even for more advanced models**.
>
> We plan to complete all other experiments and incorporate these additional results into our paper.
>
> | Dataset | GPT-3.5 (0-shot) | DP-ICL ($\varepsilon$ = 3) | DP-ICL ($\varepsilon$ = 8) | GPT-3.5 (4-shot agg.) | GPT-3.5 (4-shot) |
> |---------|------------------|------------------|----------------|----------------|----------------|
> | AGNews  | 76.20            |      86.08    |     86.18    | 85.60       |  84.40         |
> | TREC    | 77.60               |   82.57        | 82.85      | 83.00        | 81.60         |
>
>
>
> Thanks again for the valuable feedback!

---

### Official Review · Reviewer_cEpf · 2023-11-10

**Soundness:** 3 good
**Presentation:** 3 good
**Contribution:** 3 good
**Rating:** 6
**Confidence:** 2

**Summary:**

This paper tackles the problem of differential privacy in the context of in-context learning. To this end, the authors propose a framework that consists of partitioning, predicting, and aggregating. For the aggregation, the authors propose several algorithms depending on the scenarios: private voting for text classification & embedding space aggregation (ESA), and keyword space aggregation (KSA) for language generations. Through the experiments on 4 text classification datasets, 1 QA dataset, and 1 summarization dataset, the authors demonstrate the proposed method successfully imposes differential privacy without the loss of accuracy, compared to the non-private counterpart.

**Strengths:**

1. **Well-motivated problem.** Differential privacy of LLMs is an interesting and important problem.

**Weaknesses:**

1. **Rooms for the improvement of the draft.** First, lots of details are currently missing. For example, there is no explanation of how $\epsilon$ determines the noise $\sigma$. While there is no explicit mention in the main draft, the results in Appendix B (Theorems 3 and 4) might be used. But, even if this is true, there is no mention of how $\delta$ is set. In addition, how the given # of queries (e.g., 10,000 queries in Sec 4.1) is used for the algorithm? In Algorithms 1~4, there is no relevant part.
2. **Limited technical novelty.** While the introduction of differential privacy in the context of ICL is novel, most of the techniques are adopted from the previous works.
3. **Limited motivation for ESA.** The authors propose both ESA and KSA for language generation tasks. However, it seems that there is no merit in using ESA over KSA, as KSA always outperforms ESA in the experiments. Since introducing several candidates even makes practitioner confused about what to use, more support for ESA are required. In addition, it would be good to include qualitative examples of ESA’s sentence candidates with 0-shot inference and the finally selected one.

**Questions:**

Please address the concerns in above.

---

> ### Author Response · Authors · 2023-11-17
>
> We thank the reviewer for the thoughtful and constructive feedback. We would like to address the concerns highlighted in your review:
>
> > **Q1. [Details for privacy accounting, privacy parameter, and algorithms]** *“... explanation of how eps determines the noise sigma … how delta is set. In addition, how the given # of queries (e.g., 10,000 queries in Sec 4.1) is used for the algorithm?”*
>
> **A:**
>
> **1. Privacy accounting approaches (i.e., how $\varepsilon, \delta$ determines noise)**:
> - **For the experiments on text classification and embedding space aggregation (ESA)**, we use Privacy Loss Random Variable (PRV) accountant [1] because we are composing (subsampled) Gaussian mechanisms. The modern PRV-based privacy accountant computes the privacy loss numerically, and hence the final privacy loss has no closed-form expressions. We stress that PRV accountant technique is a standard practice in differential privacy literature nowadays due to the tight privacy bound it can compute. We detailed the use of the PRV accountant in the last paragraph of Appendix B.2 of our submission. To further highlight this paragraph, **we have converted that paragraph into Remark 1, and we have added a small paragraph in the main text to stress the use of PRV accountant and point the reader to Remark 1**.
> - **For the experiments on keyword space aggregation (KSA)**, we are composing (Subsampled) Propose-Test-Release (PTR) paradigm and Exponential Mechanism (EM). Since the PRV is unknown for either Propose-Test-Release or EM, PRV accountant is not applicable here and we instead use the tools of *Renyi DP* and *approximate Renyi DP* for privacy accounting. Similar to PRV-based accountant, RDP-based accountant also compute the privacy loss numerically, and hence the final privacy loss has no closed-form expressions. The detailed (approximate) RDP analysis for the PTR paradigm and Joint EM can be found in Theorem 9 to 12. **We have added a small paragraph in the main text to stress the use of RDP accountant and point the reader to Appendix B.3**.
>
> **2. Choice of $\delta$**: We thank the reviewer for pointing out that $\delta$ is missing in the main context. This is detailed in Appendix D, and **we have now highlighted it in the main text** for the reader’s convenience. Following the common practice in the literature, throughout the paper, $\delta$ is set to be $\approx 1/n$ where $n$ is the total data count. That is, $\delta=10^{-4}$ for classification task, $\delta=4 \times 10^{-6}$ for document QA task, and $\delta=5 \times 10^{-5}$ for dialog summarizations task.
>
> **3. How is the given # of queries used for the algorithm?**: The DP algorithms we develop in Section 3 are to privatize the ICL’s output *per query*. Similar to other applications of DP mechanism, for each received query, we execute the DP mechanism once. **We have added a meta algorithm at the beginning of Appendix A (Algorithm 1) to make this point clear**. The final privacy parameter is computed based on the total number of queries through the privacy accounting techniques.
>
> We thank the reviewer for the valuable questions and **have revised and incorporated these details into the paper**. If there’s anything that is not clear, we are more than happy to provide more detailed elaboration!
>
> [1] Gopi, et al. Numerical composition of differential privacy. NeurIPS 2021

---

> ### Author Response · Authors · 2023-11-17
>
> > **Q2. [Novelty of the proposed techniques]**
>
> **A:** We humbly disagree with the comment that our work has limited novelty. While privatization techniques such as “sample-and-aggregate” paradigm and Gaussian mechanism used in our paper are standard techniques in DP, these are just one facet of our paper's content. Here, we outline the major technical difficulties tackled in our paper:
>
> **Innovation in algorithm design:** The major technical challenge we faced during our research was the privatization of the language generation task, due to the potentially infinitely large output space. **How to maintain the utility of the privately aggregated sentences while safeguarding the privacy guarantee, for this setting, is highly non-trivial.** To our best knowledge, similar tasks (even non-private sentence ensemble) have never been explored in any previous work. To tackle such a challenge, our high-level idea is first to map the generated sentences into a lower-dimensional space, perform private aggregation, and then map the aggregated results back to the sentence space.
> - For ESA, while it might be natural to think of using a publicly available text-to-embedding model, mapping the aggregated embedding back to the fluent sentence is an unexplored research area. In our paper, we instead search over the sentences generated from the zero-shot prompt (which does not use the privacy budget) and output the generated sentence that is the closest to the aggregated embedding vector.
> - The development of the KSA method is also far from trivial. We came up with a novel and elegant technique where we leverage the power of LLM and reconstruct a complete sentence based on a set of keywords. Therefore, we reduce the problem of privately aggregating generated sentences to the problem of finding the high-frequency keywords among the generated sentences in a differentially private way, where we can leverage the recent advances in differentially private top-K selection.
>
> Overall, **both ESA and KSA are not simple extensions of existing DP paradigms** but, as other reviewers commented, are novel (Reviewer 3Mai, ZNqV) and suitable solutions to intricate problems of differentially private QA and text summarization (Reviewer 2dnr). That is, while we reduce the problem to more manageable spaces and adopt mainstream techniques such as the Gaussian mechanism for privatization, **the reduction itself is highly non-trivial**.
>
> Furthermore, while the resulting DP algorithm may seem simple, we stress that simplicity is appreciated in DP as **complex mechanisms pose challenges for correct implementation and auditing** [1, 2]. We believe all DP problems should have simple solutions; we are proud to contribute such a solution to the domain of in-context learning.
>
> **Innovation in privacy analysis:** In Appendix B.3.3, we derive the bound of privacy amplification by subsampling for approximate RDP. To the best of our knowledge, **our work is the first to derive this theoretical result**. We did not highlight this in the main text in order to not interrupt the overall storyline. We have now mentioned this in the main text for its higher visibility.
>
>
> [1] Lyu et al. Understanding the Sparse Vector Technique for Differential Privacy. VLDB 2017
>
> [2] Gaboardi et al. A programming framework for OpenDP. Manuscript, May (2020).

---

> ### Author Response · Authors · 2023-11-17
>
> > **Q3-1 [Motivation for ESA]** *“The authors … KSA always outperforms ESA in the experiments.”*
>
> **A:** We would like to point out that KSA does **not** always outperform ESA in our experiments. In Table 4, when comparing ESA and KSA by PTR on ROUGE-2/ROUGE-L scores, ESA constantly outperformed KSA by PTR.
> **Reason:** For the ROUGE-1 metric, it measures the 1-gram loss. KSA provides the LLM with a prompt consisting of the keywords, meaning that the LLM is likely to produce the keywords, in turn minimizing the 1-gram loss. This subtle bias in terms of one metric is one of the challenges in designing NLP methods, and it is why we provide multiple methods because we anticipate that the tradeoffs of each method are best understood by ML practitioners who will implement them.
>
> In addition, KSA with joint EM is not suitable for extremely large or infinite output domains, and ESA is capable of handling such cases.
>
> Furthermore, ESA technique also has the following advantages:
>
> **(1) Ease of Understanding and Implementation.** ESA technique is easy to understand and implement, which makes it accessible for ML practitioners. We stress that ease of implementation stands as a significant advantage in the realm of differential privacy, as it enables easier verification of the privacy analysis. Many notorious errors in the DP community have arisen as a result of the complexity of implementing DP methods [1, 2, 3]. Hence, simplicity is appreciated in the field of DP and easy-to-implement algorithms are ultimately more likely to make a positive impact in the real world.
>
> **(2) Potential for Future Enhancements.** While ESA may currently be outperformed by KSA in some aspects, it holds significant potential for future improvements. For example, after our submission, a new work [4] shows that it is possible to reconstruct the sentence directly from semantic sentence embeddings. Therefore, we could expect future embedding reconstruction methods to further improve performance.
>
>
> > **Q3-2 [Qualitative examples for ESA’s sentence candidates with 0-shot inference and the finally selected one]**
>
> **A:** Thanks for your suggestion! We have now included the qualitative examples of ESA’s sentence candidates and the finally selected one in **Appendix G**. We observe the final selected candidate is close to the ground truth compared to other candidates.
>
> We hope our response could address your concerns. We look forward to your further comments and suggestions.
>
> [1] Tramèr et al. Debugging Differential Privacy: A Case Study for Privacy Auditing. ArXiv 2022.
>
> [2] Carlini et al. No Free Lunch in "Privacy for Free: How does Dataset Condensation Help Privacy. ArXiv 2022.
>
> [3] Hayes et al. Bounding Training Data Reconstruction in DP-SGD. ArXiv 2023.
>
> [4] Morris et al. Text Embeddings Reveal (Almost) As Much As Text. ArXiv 2023.

---

> ### Author Response · Authors · 2023-11-21
> **Looking forward to hearing from you**
>
> Dear Reviewer cEpf,
>
> We deeply appreciate the time and effort you have invested in reviewing our paper. We hope we have addressed all your concerns appropriately, and we look forward to further discussion regarding any questions or comments you may have.
>
> Best,
>
> Authors

---

> > ### Comment · Reviewer_cEpf · 2023-11-21
> >
> > Thank you very much for the response. I appreciate the effort that the authors put into addressing my questions. I believe that the above results and discussion significantly improve the quality of the manuscript. My major concerns are now addressed; hence I raise my score accordingly.

---

> > > ### Author Response · Authors · 2023-11-21
> > > **Thank you**
> > >
> > > We are very glad that our response addresses all your concerns. Thank you again for helping us to improve our paper.

---

### Author Response · Authors · 2023-11-17
**General response to all reviewers**

We thank all reviewers for providing helpful feedback and suggestions. We considered the reviews carefully and modified our paper accordingly. All modifications are highlighted in red. Here’s a summary of our major revision to the paper:

- **Enhanced Clarity**: The main paper now includes additional details and guidelines regarding privacy parameter setup as well as the location of privacy analysis in the Appendix.
- **Additional Experiments on GPT3.5 and GPT4**: We have conducted additional experiments evaluating the zero-shot and four-shot performance of GPT3.5 and GPT4, detailed in Appendices E.3 and F.3, respectively.
- **Additional Experiments of few-shot learning with public dataset**: A new set of experiments focusing on four-shot predictions using public data is presented in Appendix E.2.
- **Additional Plots Showing Cost-Accuracy Trade-off**: Figures of the trade-off between cost and accuracy are presented in Appendices E.5 and F.4.
- **Examples of ESA's output**: A new section (Appendix G) included qualitative examples of ESA outputs.

---

### Meta-Review · Area_Chair_6Hav · 2023-12-26

**Metareview:**

The paper explores private methods for recently emergent in-context learning phenomenon in Large Language Models (LLMs). The paper addresses a critical and timely issue in the field of LLMs. It introduces Differentially Private In-context Learning (DP-ICL), a framework for generating private responses through consensus among ensemble LLM responses based on disjoint exemplar sets. Experiments are conducted for both classification and generation showing utility-privacy tradeoffs. Reviewers had a lukewarm response to the paper initially with concerns raised about presentation clarity, novelty, metrics, and limited practical applicability due to constraints on the number of queries. We thank both the authors and reviewers to actively engage in the discussion period. Some of the concerns of the reviewers were resolved by author response and the extra experiments on GPT-3/4. Please include these additional experiments in the main paper and also add discussion on why DP-ICL is better than vanilla ICL on GPT-3.5 experiments.

**Justification For Why Not Higher Score:**

Clarity: Lack of detail at various places
Incremental novelty: Heavy reliance on existing techniques
Practical relevance: The constraint in the number of queries and the necessity of DP-ICL

**Justification For Why Not Lower Score:**

Significance of Topic: The paper tackles privacy in in-context learning, which is very timely and of interest to the community.
Comprehensive Empirical Results: Demonstrated effectiveness of the approach on both classification and generation tasks.

---

### Decision · Program_Chairs · 2024-01-16

Accept (poster)